# FAME: Formal Abstract Minimal Explanation for Neural Networks

**Ryma Boumazouza**[*1,2], **Raya Elsaleh**[*3], **Melanie Ducoffe**[1,2], **Shahaf Bassan**[3] **and Guy Katz**[3]

[1]Airbus SAS, France, [2]IRT Saint-Exupery, France, [3]The Hebrew University of Jerusalem, Israel

## ABSTRACT

We propose **FAME** (Formal Abstract Minimal Explanations), a new class of abductive explanations grounded in abstract interpretation. FAME is the first method to scale to large neural networks while reducing explanation size. Our main contribution is the design of dedicated perturbation domains that eliminate the need for traversal order. FAME progressively shrinks these domains and leverages LiRPA-based bounds to discard irrelevant features, ultimately converging to a **formal abstract minimal explanation**. To assess explanation quality, we introduce a procedure that measures the worst-case distance between an abstract minimal explanation and a true minimal explanation. This procedure combines adversarial attacks with an optional VERIX+ refinement step. We benchmark FAME against VERIX+ and demonstrate consistent gains in both explanation size and runtime on medium- to large-scale neural networks.

## 1 INTRODUCTION

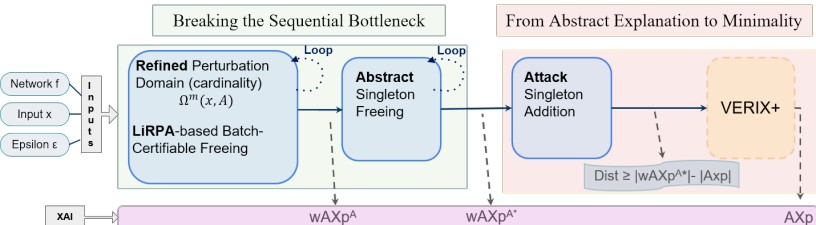

Figure 1: **FAME Framework.** The pipeline operates in two main phases **(1) Abstract Pruning (Green)** phase leverages abstract interpretation (LiRPA) to simultaneously free a large number of irrelevant (pixels that are certified to have no influence on the model's decision) features based on a batch certificate (Section 4.2). This iterative process operates within a refined, cardinality-constrained perturbation domain, $\Omega^m(x, A)$ (Eq. 5) to progressively tighten the domain; To ensure that the final explanation is as small as possible, the remaining features that could not be freed in batches are tested individually (Section 5). **(2) Exact Refinement (Orange)** phase identifies the final necessary features using singleton addition attacks and, if needed, a final run of VERIX+ (Section 6). The difference in size, $|\text{wAXp}^{A^\star}| - |\text{AXp}|$, serves as an evaluation metric of phase 1.

Neural network-based systems are being applied across a wide range of domains. Given AI tools' strong capabilities in complex analytical tasks, a significant portion of these applications now involves tasks that require reasoning. These tools often achieve impressive results in problems requiring intricate analysis to reach correct conclusions. Despite these successes, a critical challenge remains: understanding the reasoning behind neural network decisions. The internal logic of a neural network is often opaque, with its conclusions presented without accompanying justifications. This lack of transparency undermines the trustworthiness and reliability of neural networks, especially in high-stakes or regulated environments. Consequently, the need for interpretable and explainable AI (XAI) has become a growing focus in recent research.

---

[*]Equal contribution.

Two main approaches have emerged to address this challenge. The first employs statistical and heuristic techniques to infer explanations based on network's internal representations (Fel et al., 2022). While these methods estimate feature importance, that require empirical evaluation (such as the $\mu$-Fidelity metric (Bhatt et al., 2020)), the second approach leverages automated reasoners and formal verification to provide provably correct explanations grounded in logical reasoning.

We ground our work in the formal definition of Abductive Explanations (AXp) (Ignatiev et al., 2019), a concept belonging to the broader family of "formal XAI" which includes minimal explanations, also known as local-minimal, minimal unsatisfiable subsets (Marques-Silva, 2010) and prime implicants (Shih et al., 2018). An AXp is a subset of features guaranteed to maintain the model's prediction under any perturbation within a defined domain. In a machine learning context, these explanations characterize feature sets where removing any single feature invalidates the guarantee, effectively representing subsets that preserve the decision's robustness. However, a major hurdle for formal XAI is its high computational cost due to the complexity of reasoning, preventing it from scaling to large neural networks (NNs) (Marques-Silva, 2023b). This limitation, combined with the scarcity of open-source libraries, significantly hinders its adoption. Initial hybrid approaches, such as the EVA method (Fel et al., 2023), have attempted to combine formal and statistical methods, but these often fail to preserve the mathematical properties of the explanation. However, robustness-based approaches address the scalability challenges of formal XAI for NN by leveraging a fundamental connection between AXps and adversarial examples (Huang & Marques-Silva, 2023).

In this work, we present FAME, a scalable framework for formal XAI that addresses the core limitations of existing methods. Our contributions are fourfold:

- **Formal abstract explanations.** We introduce the first class of abductive explanations derived from abstract interpretation, enabling explanation algorithms to handle high-dimensional NNs.

- **Eliminating traversal order.** We design perturbation domains and a recursive refinement procedure that leverage Linear Relaxation based Perturbation Analysis (LiRPA)-based certificates to simultaneously discard multiple irrelevant features. This removes the sequential bottleneck inherent in prior work and yields an abstract minimal explanation.

- **Provable quality guarantees.** We provide the first procedure to measure the worst-case gap between abstract minimal explanations and true minimal abductive explanations, combining adversarial search with optional VERIX+ refinement.

- **Scalable evaluation.** We benchmark FAME on medium- and large-scale neural networks, showing consistent improvements in both explanation size and runtime over VERIX+. Notably, we produce the first abstract formal abductive explanations for a ResNet architecture on CIFAR-10, demonstrating scalability where exact methods become intractable.

## 2 ABDUCTIVE EXPLANATIONS & VERIFICATION

### 2.1 NOTATIONS

Scalars are denoted by lower-case letters (e.g., $x$), and the set of real numbers by $\mathbb{R}$. Vectors are denoted by bold lower-case letters (e.g., $\mathbf{x}$), and matrices by upper-case letters (e.g., $W$). The $i$-th component of a vector $\mathbf{x}$ (resp. line of a matrix $W$) is written as $\mathbf{x}_i$ (resp. $W_i$). The matrix $W^{\geq 0}$ (resp. $W^{\leq 0}$) represents the same matrix with only nonnegative (resp. nonpositive) weights. Sets are written in calligraphic font (e.g., $\mathcal{S}$). We denote the perturbation domain by $\Omega$ and the property to be verified by P.

### 2.2 THE VERIFICATION CONTEXT

We consider a neural network as a function $f : \mathbb{R}^n \to \mathbb{R}^k$. The core task of verification is to determine whether the network's output $f(x')$ satisfies a given property P for every possible input $x'$ within a specified domain $\Omega(x) \subseteq \mathbb{R}^n$. When verification fails, it means there is at least one input $x'$ in the domain $\Omega(x)$ that violates the property P (a counterexample). The verification task can be written as: $\forall x' \in \Omega(x)$, does $f(x')$ satisfy P? This requires defining two components:

1. **The Perturbation Domain** ($\Omega$): This domain defines the set of perturbations. It is often an $l_p$-norm ball around a nominal input $x$, such as an $l_\infty$ ball for modeling imperceptible noise: $\Omega = \{\mathbf{x}' \in \mathbb{R}^n \mid \|\mathbf{x} - \mathbf{x}'\|_\infty \leq \epsilon\}$.

2. **The Property** (P): This is the specification the network must satisfy. For a classification task where the network correctly classifies an input $\mathbf{x}$ into class $c$, the standard robustness property P asserts that the logit for class $c$ remains the highest for any perturbed input $\mathbf{x}'$:

$$\text{P}(\mathbf{x}') \equiv \min_{i \neq c} \{f_c(\mathbf{x}') - f_i(\mathbf{x}')\} > 0 \tag{1}$$

For instance, given an MNIST image $x$ of a '7' and a perturbation radius $\epsilon$, the property $P$ holds if the network's logit for class '7' provably exceeds all other logits for every perturbed image $x' \in \Omega(x)$.

A large body of work has investigated formal verification of NNs, with adversarial robustness being the most widely studied property (Urban & Miné, 2021). Numerous verification tools are now available off-the-shelf, and for piecewise-linear models $f$ with corresponding input domains and properties, exact verification is possible (Katz et al., 2017; Botoeva et al., 2020). In practice, however, exact methods quickly become intractable for realistic networks. To address this, we rely on Abstract Interpretation, a theory of sound approximation. Specifically, we utilize Linear Relaxation-based Perturbation Analysis (LiRPA) (Zhang et al., 2018; Singh et al., 2019) which efficiently over-approximates the network's output by enclosing it between linear upper and lower bounds. Such abstractions enable sound but conservative verification: if the relaxed property holds, the original one is guaranteed to hold as well. We provide a comprehensive background in Appendix A.

### 2.3 ABDUCTIVE EXPLANATIONS: PINPOINTING THE "WHY"

**Understanding Model Robustness with Formal Explanations:** Neural networks often exhibit sensitivity to minor input perturbations, a phenomenon that certified training can mitigate but not eliminate (De Palma et al., 2025). Even robustly trained models may only have provably safe regions spanning a few pixels for complex tasks like ImageNet classification (Serrurier et al., 2021). To build more reliable systems, it is crucial to understand *why* a model's prediction is robust (or not) within a given context. Formal explainability provides a rigorous framework for this analysis.

We focus on *abductive explanations* (AXps, also called distance-restricted explanations ($\epsilon$-AXp)) (Ignatiev et al., 2019; Huang & Marques-Silva, 2023), which identify a subset of input features that are *sufficient* to guarantee that the property P holds. Formally, a local formal abductive explanation is defined as a subset of input features that, if collapsed to their nominal values (i.e., the sample $\mathbf{x}$), ensure that the local perturbation domain $\Omega$ surrounding the sample contains no counterexamples.

**Definition 2.1** (Weak Abductive Explanation (wAXp) ). Formally, given a triple $(\mathbf{x}, \Omega, \text{P})$, an *explanation* is a subset of feature indices $\mathcal{X} \subseteq \mathcal{F} = \{1, \dots, n\}$ such that

$$\text{wAXp: } \forall \mathbf{x}' \in \Omega(\mathbf{x}), \quad \left(\bigwedge_{i \in \mathcal{X}} (\mathbf{x}'_i = \mathbf{x}_i)\right) \implies f(\mathbf{x}') \models \text{P}. \tag{2}$$

While many such explanations may exist (the set of all features $\mathcal{F}$ is a trivial one), the most useful explanations are the most concise ones (Bassan & Katz, 2023). We distinguish three levels:

**Minimal Explanation:** An explanation $\mathcal{X}$ is *minimal* if removing any single feature from it would break the guarantee (i.e., $\mathcal{X} \setminus \{j\}$ is no longer an explanation for any $j \in \mathcal{X}$). These are also known as minimal unsatisfiable subsets(Ignatiev et al., 2016; Bassan & Katz, 2023).

**Minimum Explanation:** An explanation $\mathcal{X}$ is *minimum* if it has the smallest possible number of features (cardinality) among all possible minimal explanations.

The concept of an abductive explanation is illustrated using a classification task (details in Appendix D.1, Figure 4). The goal is to find a minimal subset of fixed features ($\mathcal{X}$) that guarantees a sample's classification within its perturbation domain. For the analyzed sample, fixing $\mathbf{x}_2$ alone is insufficient due to the existence of a counterexample (Figure 5). However, fixing the set $\mathcal{X} = \{\mathbf{x}_2, \mathbf{x}_3\}$ creates a 'safe' subdomain without counterexamples, confirming it is an abductive explanation. This explanation is minimal (neither $\mathbf{x}_2$ nor $\mathbf{x}_3$ work alone) but not minimum in cardinality, as $\mathcal{X}' = \{\mathbf{x}_1\}$ is also a valid minimal explanation. In the rest of this paper, we will use the terms abductive explanation or formal explanation and the notation wAXp to refer to Definition 2.1.

## 3  RELATED WORK

Substantial progress has been made in the practical efficiency of computing formal explanations. While finding an abductive explanation (AXp) is tractable for some classifiers (Marques-Silva, 2023a; Darwiche & Ji, 2022; Huang et al., 2022; 2021; Izza et al., 2020; Marques-Silva et al., 2020; 2021), it becomes computationally hard for complex models like random forests and neural networks (Ignatiev & Marques-Silva, 2021; Izza & Marques-Silva, 2021). To address this inherent complexity, these methods typically encode the problem as a logical formula, leveraging automated reasoners like SAT, SMT, and Mixed Integer Linear Programming (MILP) solvers (Audemard et al., 2022; Ignatiev, 2020; Ignatiev et al., 2022; Ignatiev & Marques-Silva, 2021; Izza & Marques-Silva, 2021) . Early approaches, such as deletion-based (Chinneck & Dravnieks, 1991) and insertion-based (de Siqueira, 1988) algorithms, are inherently sequential, thus requiring an ordering of the input features traditionally denoted as *traversal ordering*. They require a number of costly verification calls linear with the number of features, which prevents effective parallelization. As an alternative, surrogate models have been used to compute formal explanations for complex models (Boumazouza et al., 2021; 2023), but the guarantee does not necessary hold on the original model.

Recent work aims to break the sequential bottleneck, by linking explainability to adversarial robustness and formal verification. DistanceAXp (Huang & Marques-Silva, 2023; La Malfa et al., 2021) is a key example, aligning with our definition of AXp and enabling the use of verification tools.

The latest literature focuses on breaking the sequential bottleneck using several strategies that include parallelization. This is achieved either by looking for several counterexamples at once (Izza et al., 2024; Bassan & Katz, 2023; La Malfa et al., 2021; Bassan et al., 2023) or by identifying a set of irrelevant features simultaneously, as seen in VERIX (Wu et al., 2023), VERIX+ (Wu et al., 2024b), and prior work (Bassan & Katz, 2023). For instance, VERIX+ introduced stronger traversal strategies to alleviate the sequential bottleneck. Their binary search approach splits the remaining feature set and searches for batches of consecutive irrelevant features, yielding the same result as sequential deletion but with fewer solver calls. They also adapted QuickXplain (Junker, 2004), which can produce even smaller explanations at the cost of additional runtime by verifying both halves. Concurrently, (Bassan & Katz, 2023) proposed strategies like the singleton heuristic to reuse verification results and derived provable size bounds, but their approach remains significantly slower than VERIX+ and lacks publicly available code.

The identified limitations are twofold. First, existing methods rely heavily on exact solvers such as Marabou (Katz et al., 2019; Wu et al., 2024a), which do not scale to large NNs and are restricted to CPU execution. Recent verification benchmarks (Brix et al., 2023; Ducoffe et al., 2024; Zhao et al., 2022) consistently demonstrate that GPU acceleration and distributed verification are indispensable for achieving scalability. Second, these approaches critically depend on traversal order. As shown in VERIX, the chosen order of feature traversal strongly impacts both explanation size and runtime. Yet, determining an effective order requires prior knowledge of feature importance, precisely the information that explanations are meant to uncover, thus introducing a circular dependency. Nevertheless, VERIX+ currently represents the SOTA for abductive explanations in NNs, achieving the best trade-off between explanation size and computation time.

Our work builds on this foundation by directly addressing the sequential bottleneck of formal explanation without requiring a traversal order, a first in formal XAI. We demonstrate that leveraging incomplete verification methods and GPU hardware is essential for practical scalability. Our approach offers a new solution to the core scalability issues, complementing other methods that aim to reduce explanation cost through different means (Bassan et al., 2025b;a).

## 4  FAME: FORMAL ABSTRACT MINIMAL EXPLANATION

In this section, we introduce FAME, a framework that builds *abstract abductive explanations* (Definition 4.1). FAME proposes novel strategies to provide sound abstract abductive explanations (wAXp$^A$) such as an Abstract Batch Certificate using Knapsack formulation, and a Recursive Refinement, relying on raw bounds provided by a formal framework (we use LiRPA in this paper).

**Definition 4.1** (Abstract Abductive Explanation (wAXp$^A$)). Formally, given a triple $(\mathbf{x}, \Omega, \mathrm{P})$, an *abstract abductive explanation* is a subset of feature indices $\mathcal{X}^A \subseteq \mathcal{F} = \{1, \dots, n\}$ such that, under

an abstract interpretation $\overline{f}$ of the model $f$, the following holds:

$$\text{wAXp}^A : \forall \mathbf{x}' \in \Omega(\mathbf{x}), \quad \left( \bigwedge_{i \in \mathcal{X}^A} (\mathbf{x}'_i = \mathbf{x}_i) \right) \implies \overline{f}(\mathbf{x}') \models \text{P}. \tag{3}$$

Here, $\overline{f} = \text{LiRPA}(f, \Omega)$ denotes the sound over-approximated bounds of the model outputs on the domain $\Omega$, as computed by the LiRPA method. If Eq. (3) holds, any feature outside $\mathcal{X}^A$ can be considered irrelevant *with respect to the abstract domain*. This ensures that the concrete implication $f(\mathbf{x}') \models \text{P}$ also holds for all $x' \in \Omega$. In line with the concept of abductive explanations, we define an *abstract minimal explanation* as an abstract abductive explanation ($\text{wAXp}^{A^\star}$) a set of features $\mathcal{X}^A$ from which no feature can be removed without violating Eq. (3).

Due to the over-approximation, as detailed in Section 2.2, any *abstract abductive explanation* is a *weak abductive explanation* for the model $f$. In the following we present the first steps described in Figure 1 to build such a $\text{wAXp}^A$.

## 4.1 THE ASYMMETRY OF PARALLEL FEATURE SELECTION

In the context of formal explanations, **adding a feature** means identifying it as essential to a model's decision (causes the model to violate the desired property P), so its value must be fixed in the explanation. Conversely, **freeing a feature** means identifying it as irrelevant, allowing it to vary without affecting the prediction. A key insight is the asymmetry between these two actions: while adding necessary features can be parallelized naively, freeing features cannot due to complex interactions.

**Proposition 4.1 (Simultaneous Freeing).** it is unsound to free multiple features at once based only on individual verification as two features may be individually irrelevant yet jointly critical.

Parallelizing feature freeing based on individual verification queries is unsound due to hidden feature dependencies that stem from treating the verifier as a simple binary oracle (SAT/UNSAT; see Appendix A for formal definitions) (Proposition 4.1). To solve this, we introduce the Abstract Batch Certificate $\Phi(\mathcal{A})$ (Definition 4.2). Unlike naive binary checks, $\Phi(\mathcal{A})$ leverages abstract interpretation to compute a joint upper bound on the worst-case contribution of the entire set $\mathcal{A}$ simultaneously. If $\Phi(\mathcal{A}) \leq 0$, it mathematically guarantees that simultaneously freeing $\mathcal{A}$ is sound, explicitly accounting for their combined interactions. The formal propositions detailing this asymmetry is provided in the Appendix B.

## 4.2 ABSTRACT INTERPRETATION FOR SIMULTANEOUS FREEING

Standard solvers act as a "binary oracle" and their outcomes (SAT/UNSAT) are insufficient to certify batches of features for freeing without a traversal order. This is because of feature dependencies and the nature of the verification process. We address this by leveraging *inexact* verifiers based on abstract interpretation (LiRPA) to extract *proof objects* (linear bounds) that conservatively track the contribution of any feature set. Specifically, we use CROWN (Zhang et al., 2018) to define an *abstract batch certificate* $\Phi$ in Definition 4.2. If one succeeds in freeing a set of features $\mathcal{A}$ given $\Phi$, we denote such an explanation as a *formal abstract explanation* that satisfies Proposition 4.2.

**Definition 4.2 (Abstract Batch Certificate).** Let $\mathcal{A}$ be a set of features and $\Omega$ any perturbation domain. The *abstract batch certificate* is defined as:

$$\Phi(\mathcal{A}; \Omega) = \max_{i \neq c} \left( \overline{b}^i(x) + \sum_{j \in \mathcal{A}} c_{i,j} \right),$$

where the baseline bias (worst-case margin of the model's output) at $x$ is $\overline{b}^i(x) = \overline{W}^i \cdot x + \overline{w}^i$,

and the contribution of each feature $j \in \mathcal{A}$ is $c_{i,j} = \max \left\{ \overline{W}_j^{i,\geq 0} (\overline{x}_j - x_j), \ \overline{W}_j^{i,\leq 0} (\underline{x}_j - x_j) \right\}$,

with $\overline{x}_j = \max\{x'_j : x' \in \Omega(x)\}$ and $\underline{x}_j = \min\{x'_j : x' \in \Omega(x)\}$. The weights $\overline{W}^i$ and biases $\overline{w}^i$ are obtained from LiRPA bounds, which guarantee for each target class $i \neq c$, with $c$ being the groundtruth class:

$$\forall x' \in \Omega(x), \quad f_i(x') - f_c(x') \leq \overline{f}_{i,c}(x') = \overline{W}^i \cdot x' + \overline{w}^i,$$

**Proposition 4.2** (**Batch-Certifiable Freeing**)**.** If $\Phi(\mathcal{A}; \Omega) \leq 0$, then $\mathcal{F} \setminus \mathcal{A}$ is a weak abductive explanation (wAXp).

**Lemma 4.1.** If $\Phi(\mathcal{A}) \leq 0$, freeing all features in $\mathcal{A}$ is sound; that is, the property P holds for every $x' \in \Omega(x)$ with $\{x'_k = x_k\}_{k \in \mathcal{F} \setminus \mathcal{A}}$.

The proof of Proposition 4.2 is given in Appendix B. The trivial case $\mathcal{A} = \emptyset$ always satisfies the certificate, but our goal is to efficiently certify large feature sets. The abstract batch certificate also highlights two extreme scenarii. In the first, if $\Phi(\mathcal{F}) \leq 0$, all features are irrelevant, meaning the property P holds across $\Omega$ without fixing any inputs. In the second, if $\overline{b}^i(x) \geq 0$ for some $i \neq c$, then $\Phi(\emptyset) > 0$ and no feature can be safely freed; this situation arises when the abstract relaxation is too loose, producing vacuous bounds. Avoiding this degenerate case requires careful selection of the *perturbation domain*, a consideration we highlight for the first time in the context of abductive explanations. The choice of abstract domain is discussed in Section 5.

### 4.3 Minimizing the Size of an Abstract Explanation via a Knapsack Formulation

Between the trivial and degenerate cases lies the nontrivial setting: finding a maximal *set of irrelevant features* $\mathcal{A}$ to free given the abstract batch certificate $\Phi$. Let $\mathcal{F}$ denote the index set of features. Maximizing $|\mathcal{A}|$ can be naturally formulated as a $0/1$ Multidimensional Knapsack Problem (MKP). For each feature $j \in \mathcal{F}$, we introduce a binary decision variable $y_j$ indicating whether the feature is selected. The optimization problem then reads:

$$\max_y \sum_{j \in \mathcal{F}} y_j \text{ s.t.} \sum_{j \in \mathcal{F}} c_{ij} y_j \leq -\overline{b}^i(x), \quad i \in I, \, i \neq c \tag{4}$$

where $c_{i,j}$ represents the contribution of feature $j$ to constraint $i$, and $-\overline{b}^i(x)$ is the corresponding knapsack capacity. The complexity of this MKP depends on the number of output classes. For binary classification ($k = 2$), the problem is linear[1]. In the standard multiclass setting ($k > 2$), however, the MKP is NP-hard. While moderately sized instances can be solved exactly using a MILP solver, this approach does not scale to large feature spaces. To ensure scalability, we propose a simple and efficient greedy heuristic, formalized in Algorithm 1. Rather than solving the full MKP, the heuristic iteratively selects the feature $j^\star$ that is *least likely* to violate any of the $k-1$ constraints, by minimizing the maximum normalized cost across all classes. An example is provided in Appendix D.2. This procedure is highly parallelizable, since all costs can be computed simultaneously. While suboptimal by design, it produces a set $\mathcal{A}$ such that $\Phi(\mathcal{A}; \Omega) \leq 0$. A key advantage of this greedy batch approach is its computational efficiency. The cost is dominated by the computation of feature contributions $c_{i,j}$. This requires a single backward pass through the abstract network, which has a complexity of $O(L \cdot N)$ (where $L$ is depth and $N$ is neurons) and is highly parallelizable on GPUs. In contrast, exact solvers require solving an NP-hard problem for each feature or batch. In Section 7, we compare the performance of this greedy approach against the optimal MILP solution, demonstrating that it achieves competitive results with dramatically improved scalability.

---

**Algorithm 1** Greedy Abstract Batch Freeing (One Step)

---
1: **Input:** model $f$, perturbation domain $\Omega^m$, candidate set $F$
2: **Initialize:** $\mathcal{A} \leftarrow \emptyset$, linear bounds $\{\overline{W}^i, \overline{w}^i\} = \text{LiRPA}(f, \Omega^m(x))$
3: **Do:** compute $c_{i,j}$ in parallel
4: **while** $\Phi(\mathcal{A}) \leq 0$ and $|\mathcal{F}| > 0$ **do**
5:     pick $j^\star = \arg\min_{j \in F \setminus \mathcal{A}} \max_{i \neq c} c_{i,j}/(-\overline{b}_i)$            ▷ Parallel reduction
6:     **if** $\Phi(\mathcal{A} \cup \{j^\star\}) \leq 0$ and $|\mathcal{A}| \leq m$ **then**
7:         $\mathcal{A} \leftarrow \mathcal{A} \cup \{j^\star\}$
8:     **end if**
9:     $F \leftarrow F \setminus \{j^\star\}$                               ▷ Remove candidate
10: **end while**
11: **Return:** $\mathcal{A}$

---

[1] it can be solved optimally in $\mathcal{O}(n)$ time by sorting features by ascending contribution $c_{1,j}$ and greedily adding them until the capacity is exhausted.

## 5 Refining the Perturbation Domain for Abductive Explanation

Previous approaches for batch freeing reduce the perturbation domain using a traversal order $\pi$, defining $\Omega_{\pi,i}(\mathbf{x}) = \{\mathbf{x}' \in \mathbb{R}^n : \|\mathbf{x} - \mathbf{x}'\|_\infty \le \epsilon, \ \mathbf{x}'_{\pi_{i:}} = \mathbf{x}_{\pi_{i:}}\}$. These methods only consider freeing dimensions up to a certain order. However, as discussed previously, determining an effective order requires prior knowledge of feature importance, the very information that explanations aim to uncover, introducing a circular dependency. This reliance stems from the combinatorial explosion: the number of possible subsets of input features grows exponentially, making naive enumeration of abstract domains intractable.

To address this, we introduce a new perturbation domain, denoted the *cardinality-constrained perturbation domain*. For instance, one can restrict to $\ell_0$-bounded perturbations:

$$\Omega^m(\mathbf{x}) = \{\mathbf{x}' \in \mathbb{R}^n : \|\mathbf{x} - \mathbf{x}'\|_\infty \le \epsilon, \ \|\mathbf{x} - \mathbf{x}'\|_0 \le m\},$$

which ensures that at most $m$ features may vary simultaneously. This concept is closely related to the $\ell_0$ norm and has been studied in verification (Xu et al., 2020), but, to the best of our knowledge, it is applied here for the first time in the context of abductive explanations. The greedy procedure in Algorithm 1 can then certify a batch of irrelevant features $\mathcal{A}$ under this domain. Once a set $\mathcal{A}$ is freed, the feasible perturbation domain becomes strictly smaller, enabling tighter bounds and the identification of additional irrelevant features. We formalize this as the *refined abstract domain* that ensures that at most $m$ features can vary in addition to the set of previously seclected ones $\mathcal{A}$:

$$\Omega^m(\mathbf{x}; \mathcal{A}) = \{\mathbf{x}' \in \mathbb{R}^n : \|\mathbf{x} - x'\|_\infty \le \epsilon, \ \|\mathbf{x}_{\mathcal{F} \setminus \mathcal{A}} - \mathbf{x}'_{\mathcal{F} \setminus \mathcal{A}}\|_0 \le m\}. \tag{5}$$

By construction, $\Omega^m(\mathbf{x}; \mathcal{A}) \subseteq \Omega^{m+|\mathcal{A}|}(\mathbf{x})$, so any free set derived from $\Omega^m(x; \mathcal{A})$ remains sound for the original budget $m + |\mathcal{A}|$. Recomputing linear bounds on this tighter domain often yields strictly smaller abstract explanation. This refinement naturally suggests a recursive strategy: after one round of greedy batch freeing, we restrict the domain to $\Omega^m(\mathbf{x}; \mathcal{A})$, recompute LiRPA bounds, and reapply Algorithm 1 for $m = 1 \ldots |\mathcal{F} \setminus \mathcal{A}|$. Unlike the static traversal of prior work (e.g., VERIX+), FAME employs a dynamic, cost-based selection by re-evaluating abstract costs $c_{i,j}$ at each recursive step. This process functions as an adaptive abstraction mechanism: iteratively enforcing cardinality constraints tightens the domain, reducing LiRPA's over-approximation error and enabling the recovery of additional freeable features initially masked by loose bounds. As detailed in Algorithm 2, this process can be iterated, progressively shrinking the domain and expanding $\mathcal{A}$. In practice, recursion terminates once no new features can be freed. Finally, any remaining candidate features can be tested individually using the binary search approach proposed by VeriX+ but replacing Marabou by CROWN (see Algorithm 5). This final step ensures that we obtain a formal abstract minimal explanation, as defined in Definition 4.1

---

**Algorithm 2** Recursive Abstract Batch Freeing

1: **Input:** model $f$, input $x$, candidate set $\mathcal{F}$
2: **Initialize:** $\mathcal{A} \leftarrow \emptyset$                                                       ▷ certified free set
3: **repeat**
4:      $\mathcal{A}_{best} \leftarrow \emptyset$
5:      **for** $m = 1 \ldots |\mathcal{F} \setminus \mathcal{A}|$ **do**
6:          $\mathcal{A}_m \leftarrow$ GREEDYABSTRACTBATCHFREEING$(f, \Omega^m(x; \mathcal{A}), \mathcal{F} \setminus \mathcal{A})$
7:          **if** $|\mathcal{A}_m| > |\mathcal{A}_{best}|$ **then**
8:              $\mathcal{A}_{best} \leftarrow \mathcal{A}_m$
9:          **end if**
10:      **end for**
11:      $\mathcal{A} \leftarrow \mathcal{A} \cup \mathcal{A}_{best}$
12: **until** $\mathcal{A}_{best} = \emptyset$
13: $\mathcal{A} =$ ITERATIVE SINGLETON FREE$(f, x, \mathcal{F}, \mathcal{A})$         ▷ refine by testing remaining features
14: **Return:** $\mathcal{A}$

---

## 6 Distance from Abstract Explanation to Minimality

Algorithm 2 returns a *minimal abstract explanation*: with respect to the chosen LiRPA relaxation, the certified free set $\mathcal{A}$ cannot be further enlarged. This guarantee is strictly weaker than minimality

in the exact sense. The remaining features may still include irrelevant coordinates that abstract interpretation fails to certify, due to the coarseness of the relaxation. In other words, minimality is relative to the verifier: stronger but more expensive verifiers (e.g., Verix+ with Marabou) are still required to converge to a *true minimal explanation*.

We achieve this via a two-phase pipeline (Figure 1). **Phase 1 (Abstract Pruning)** generates a sound abstract explanation $\text{wAXp}^{A^\star}$. **Phase 2 (Exact Refinement)** minimizes this candidate using VERIX+, ensuring the final output is guaranteed minimal. The gap arises from the tradeoff between verifier accuracy and domain size. Abstract methods become more conservative as the perturbation domain grows, while exact methods remain sound but scale poorly. This motivates hybrid strategies that combine fast but incomplete relaxations with targeted calls to exact solvers. As an additional acceleration step, adversarial attacks can be used. By Lemma B.1, if attacks identify features that must belong to the explanation, they can be added simultaneously (*see Algorithm 4*). Unlike abstract interpretation, the effectiveness of adversarial search typically increases with the domain size: larger regions make it easier to find counterexamples.

**Towards minimal explanations.** In formal XAI, fidelity is a hard constraint guaranteed by the verifier. Therefore, the explanation cardinality (minimality) becomes the only metric to compare formal abductive explanations. A smaller explanation is strictly better, provided it remains sufficient. Our strategy is to use the *minimal abstract explanation* ($\text{wAXp}^{A^\star}$) as a starting point, and then search for the closest minimal explanation. Concretely, we aim to identify the largest candidate set of potentially irrelevant features that, if freed together, would allow all remaining features to be safely added to the explanation at once. A good traversal order of the candidate space is crucial here, as it determines how efficiently such irrelevant features can be pinpointed. Formally, if $\mathcal{X}^A$ denotes the minimal abstract explanation and $\mathcal{X}^{A^\star}$ the closest minimal explanation, we define the *absolute distance to minimality* as the number of irrelevant features not captured by the abstract method: $d(\mathcal{X}^A, \mathcal{X}^{A^\star}) = \left|\mathcal{X}^A \setminus \mathcal{X}^{A^\star}\right|$.

## 7 EXPERIMENTS

To evaluate the benefits and reliability of our proposed explainability method, FAME, we performed a series of experiments comparing its performance against the SoTA VERIX+ implementation. We assessed the quality of the explanations generated by FAME by comparing them to those of VERIX+ across four distinct models, including both fully connected and convolutional neural networks (CNNs). We considered two primary performance metrics: the runtime required to compute a single explanation and the size (cardinality) of the resulting explanation.

Our experiments, as in VERIX+ (Wu et al., 2024b), were conducted on two widely-used image classification datasets: MNIST (Yann, 2010) and GTSRB (Stallkamp et al., 2012). Each score was averaged over non-robust samples from the 100 samples of each dataset. For the comparison results, the explanations were generated using the FAME framework only, and with a final run of VERIX+ to ensure minimality (See Figure 1).

| Traversal order | VERIX+ (alone) bounds | | FAME: Single-round / | | | | FAME: Iterative refinement / | | | | FAME-accelerated VERIX+ / + bounds | | |
| Search procedure | binary | | MILP | | Greedy | | MILP | | Greedy | | Greedy + binary | | |
| Metrics ↓ | $|AXp|$ | time | $|\text{wAXp}^A|$ | time | $|\text{wAXp}^A|$ | time | $|\text{wAXp}^A|$ | time | $|\text{wAXp}^A|$ | time | $\|\text{candidate-set}\|$ | $|AXp|$ | time |
|---|---|---|---|---|---|---|---|---|---|---|---|---|---|
| **MNIST-FC** | 280.16 | 13.87 | 441.05 | 4.4 | 448.37 | 0.35 | 229.73 | 14.30 | 225.14 | 8.78 | 44.21 | **224.41** | **13.72** |
| **MNIST-CNN** | 159.78 | 56.72 | 181.24 | 5.59 | 190.29 | 0.51 | 124.9 | 12.35 | 122.09 | 5.6 | 104.09 | **113.53** | **33.75** |
| **GTSRB-FC** | **313.42** | 56.18 | 236.85 | 9.68 | 243.18 | 0.97 | 331.84 | 12.28 | 332.74 | 5.26 | 11.93 | 332.66 | **9.26** |
| **GTSRB-CNN** | 338.28 | 185.03 | 372.66 | 12.45 | 379.34 | 1.35 | 322.42 | 17.63 | 322.42 | 7.42 | 219.57 | **322.42** | **138.12** |

Table 1: Average explanation size and generation time (in seconds) are compared for FAME (single-round and iterative MILP/Greedy) with FAME-accelerated VERIX+ to achieve minimality.

**Experimental Setup** All experiments were carried out on a machine equipped with an Apple M2 Pro processor and 16 GB of memory. The analysis is conducted on fully connected (-FC) and convolutional (-CNN) models from the MNIST and GTSRB datasets, with $\epsilon$ set to 0.05 and 0.01 respectively. The verified perturbation analysis was performed using the DECOMON library[2], ap-

---

[2]https://github.com/airbus/decomon

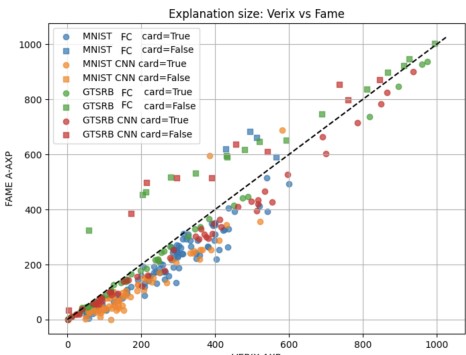 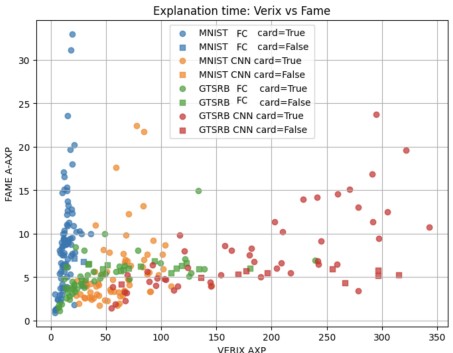

Figure 2: **FAME's iterative refinement approach against the VERIX+ baseline**. The left plot compares the size of the final explanations. The right plot compares the runtime (in *seconds*). The data points for each model are distinguished by color, and the use of circles (card=True) and squares (card=False) indicates whether a cardinality constraint ($||x - x'||_0 \leq m$) was applied.

plying the CROWN method with an $l_\infty$-norm. The NN verifier Marabou (Katz et al., 2019) is used within VERIX+. We included a sensitivity analysis covering: (1) Solver Choice, confirming the Greedy heuristic's near-optimality vs. MILP (Table 1); (2) Cardinality Constraints, showing that card=True yields significantly smaller explanations (Figure 2); and (3) Perturbation Magnitude ($\epsilon$), which we fixed to baseline used by VERIX+ for direct comparison. We include additional experimental results on the ResNet-2B architecture (CIFAR-10) from the VNN-COMP benchmark (Wang et al., 2021) to demonstrate scalability on deeper models. The complete set of hyperparameters and the detailed architectures of the models used are provided in Appendix E for full reproducibility.

## 7.1 GREEDY VS. MILP FOR ABSTRACT BATCH FREEING

**Performance in a Single Round** This experiment, in the 'FAME: Single Round' column of Table 1, compares the runtime and size of the largest free set obtained in a single round using the greedy method versus an exact MILP solver for the abstract batch freeing (Algorithm 1).

Across all models, the greedy heuristic consistently provided a significant **speedup (ranging from $9\times$ to $12\times$)** while achieving an abstract explanation size very close (fewer than 9 features in average) to that of the optimal MILP solver. This demonstrates that, for single-round batch freeing, the greedy method offers a more practical and scalable solution.

**Performance with Iterative Refinement** This experiment compares the two methods in an iterative setting of the abstract batch freeing, where the perturbation domain is progressively refined (Section 5). For the iterative refinement process, the greedy approach maintained a substantial runtime advantage over the MILP solver, with a speedup up to $2.4\times$ on the GTSRB-CNN model, while producing abstract explanations that were consistently close in size to the optimal solution. The distinction between the circle and square markers is significant in Figure 2. The square markers (card=False) tend to lie closer to or even above the diagonal line. This suggests that the cardinality-constrained domain, when successful, is highly effective at finding more compact explanations.

**Impact of Iterative Refinement:** Comparing 'FAME: Single-round' vs. 'FAME: Iterative refinement' in Table 1 isolates the impact of Algorithm 2. For MNIST-CNN, iterative refinement reduces explanation size by 36% (190.29 to 122.09). This highlights the trade-off: a modest increase in runtime yields significantly more compact explanations.

## 7.2 COMPARISON WITH STATE-OF-THE-ART (VERIX+)

We compare in this section the results of VERIX+ (alone) vs. FAME-accelerated VERIX+.

**Explanation Size and Runtime:** FAME consistently produces smaller explanations than VERIX+ while being significantly faster, mainly due to FAME's iterative refinement approach, as visually confirmed by the plots in Figure 2 that show a majority of data points falling below the diagonal line

for both size and time comparisons. The runtime gains are particularly substantial for the GTSRB models (green and red markers), where FAME's runtime is often only a small fraction of VERIX+'s as shown in Table 1. In some cases, FAME delivers a non-minimal set that is smaller than VERIX+ 's minimal set, with up to a $25\times$ speedup (322.42 features in 7.4s compared to 338.28 in 185.03s for the GTSRB-CNN model) while producing wAXp$^A$ that were consistently close in size to the optimal solution.

**The Role of Abstract Freeing:** The effectiveness of FAME's approach is further supported by the "distance to minimality" metric. The average distance to minimality was 44.21 for MNIST-FC and 104.09 for MNIST-CNN. An important observation from our experiments is that when the abstract domains in FAME are effective, they yield abstract abductive explanations wAXp$^A$ that are smaller than the abductive explanations (AXp) from VERIX+. This is not immediately obvious from the summary table, as the final explanations may differ. Conversely, when FAME's abstract domains fail to find a valid free set, our method defaults to a binary search approach similar to VERIX+. However, since we do not use the Marabou solver in this phase, the resulting wAXp$^A$ is larger than the AXp provided by Marabou. This highlights the trade-off and the hybrid nature of our approach.

Finally, to demonstrate the generality of our framework beyond standard benchmarks, in Appendix F we provide additional experiments on the ResNet-2B architecture Wang et al. (2021) trained on CIFAR-10. These results represent, to the best of our knowledge, the first formal explanations generated for such a complex architecture, highlighting FAME as an enabling technology for scalability.

## 8 CONCLUSION AND DISCUSSION

In this work, we introduced **FAME** (Formal Abstract Minimal Explanations), a novel framework for computing abductive explanations that effectively scales to large neural networks. By leveraging a hybrid strategy grounded in abstract interpretation and dedicated perturbation domains, we successfully addressed the long-standing sequential bottleneck of traditional formal explanation methods.

Our main contribution is a new approach that eliminates the need for traversal order by progressively shrinking dedicated perturbation domains and using LiRPA-based bounds to efficiently discard irrelevant features. The core of our method relies on a greedy heuristic for batch freeing that, as our analysis shows, is significantly faster than an exact MILP solver while yielding comparable explanation sizes.

Our experimental results demonstrate that the full hybrid FAME pipeline outperforms the current state-of-the-art VERIX+ baseline, providing a superior trade-off between computation time and explanation quality. We consistently observed significant reductions in runtime while producing explanations that are close to true minimality. This success highlights the feasibility of computing formal explanations for larger models and validates the effectiveness of our hybrid strategy.

Beyond its performance benefits, the FAME framework is highly generalizable. Although our evaluation focused on classification tasks, the framework can be extended to other machine learning applications, such as regression. While we focused on robustness in continuous domains, FAME's high-level algorithms (batch certificate, greedy selection) support discrete features (see Appendix B). LiRPA natively handles discrete variables (e.g., one-hot encodings) via contiguous interval bounds. Furthermore, the framework can support other properties like local stability. Additionally, FAME can be configured to use exact solvers for the final refinement step, ensuring its adaptability and robustness for various use cases.

Finally, we demonstrated FAME's scalability on the ResNet-2B (CIFAR-10) architecture. Although the abstraction gap naturally widens with depth, FAME's ability to rapidly prune irrelevant features establishes it as a critical enabling step for applying formal XAI to complex models where exact-only methods are currently intractable. By designing a framework that natively leverages certificates from modern, GPU-enabled verifiers, this work effectively bridges the gap between formal guarantees and practical scalability.

## ACKNOWLEDGEMENTS

Our work has benefited from the AI Cluster ANITI and the research program DEEL.[3] ANITI is funded by the France 2030 program under the Grant agreement n°ANR-23-IACL-0002. DEEL is an integrative program of the AI Cluster ANITI, designed and operated jointly with IRT Saint Exupéry, with the financial support from its industrial and academic partners and the France 2030 program under the Grant agreement n°ANR-10-AIRT-01. Within the DEEL program, we are especially grateful to Franck MAMALET for their constant encouragement, valuable discussions, and insightful feedback throughout the development of this work. The work of Elsaleh, Bassan, and Katz was partially funded by the European Union (ERC, VeriDeL, 101112713). Views and opinions expressed are however those of the author(s) only and do not necessarily reflect those of the European Union or the European Research Council Executive Agency. Neither the European Union nor the granting authority can be held responsible for them. The work of Elsaleh, Bassan, and Katz was additionally supported by a grant from the Israeli Science Foundation (grant number 558/24). Elsaleh is also supported by the Ariane de Rothschild Women Doctoral Program.

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
