# Appendix

The appendix collects proofs, model specifications, and supplementary experimental results that support the main paper.

**Appendix A** contains additional background on formal verification terminology, Abstract Interpretation, and LiRPA.
**Appendix B** contains the complete proofs of all propositions.
**Appendix C** provides the pseudocode for the FAME algorithms and the associated baselines.
**Appendix D** provides illustrative examples of abductive explanations and the greedy knapsack formulation.
**Appendix E** provides specifications of the datasets and architectures used, along with supplementary experimental results.
**Appendix F** details the scalability analysis on complex architectures (ResNet-2B on CIFAR-10).
**Appendix G** provides the LLM usage disclosure.

## A    BACKGROUND ON FORMAL VERIFICATION

### A.1    ABSTRACT INTERPRETATION

Abstract Interpretation is a theory of sound approximation of the semantics of computer programs. In the context of neural networks, it allows us to compute over-approximations of the network's output range without executing the network on every single point in the input domain (which is infinite).

While exact verification methods (like MILP solvers) provide precise results, they are generally NP-hard and do not scale to large networks. Abstract interpretation trades precision for scalability (typically polynomial time) by operating on abstract domains (e.g., intervals, zonotopes, or polyhedra) rather than concrete values.

### A.2    LiRPA (LINEAR RELAXATION-BASED PERTURBATION ANALYSIS)

LiRPA (Linear Relaxation-based Perturbation Analysis) is a specific, efficient instance of abstract interpretation designed for neural networks. Instead of propagating simple intervals (which become too loose/imprecise in deep networks), LiRPA propagates linear constraints. For every neuron $x_j$, it computes two linear bounds relative to the input $x$:

$$\underline{w}_j^T x + \underline{b}_j \leq f_j(x) \leq \overline{w}_j^T x + \overline{b}_j$$

These linear bounds allow us to rigorously bound the "worst-case" behavior of the network much more tightly than simple intervals. If the lower bound of the correct class minus the upper bound of the target class is positive, we have a mathematically sound certificate of robustness.

**Illustrative Example:**    Consider a nominal input image $\bar{x}$ from the MNIST dataset depicting the digit '7'. In a standard local robustness verification task, we define the input domain $\Omega(\bar{x})$ as an $l_\infty$-norm ball with a radius of $\epsilon = 0.05$. This implies that each pixel $x_i$ in the image is permitted to vary independently within the interval $[\bar{x}_i - 0.05, \bar{x}_i + 0.05]$.

The verification objective is to prove that the property $P$ holds: specifically, that for every possible perturbed image $x \in \Omega(\bar{x})$, the network's output logit for the ground-truth class ('7') remains strictly greater than the logit for any target class $k$ (e.g., '1'). In the context of LiRPA, this is verified by computing a sound lower bound for the correct class ($\underline{f}_7$) and a sound upper bound for the competing class ($\overline{f}_1$). If the verified margin $\underline{f}_7 - \overline{f}_1 > 0$, the network is guaranteed to be robust against all perturbations in $\Omega(\bar{x})$.

### A.3 VERIFICATION TERMINOLOGY

We formulate the check for explanation sufficiency as a constraint satisfaction problem. A query is SAT if a valid perturbation (counter-example) exists, and UNSAT if no such perturbation exists (meaning the explanation is valid).

- Soundness (No False Positives): A verifier is sound if it guarantees that any certified property is truly holds. In Abstract Interpretation, soundness is achieved because the computed abstract bounds strictly enclose the true concrete values. If these conservative bounds satisfy the property (UNSAT), the actual network must also satisfy it.

- Completeness (No False Negatives): A verifier is complete if it is capable of certifying *any* valid explanation. Exact solvers (like MILP) are complete. In contrast, Abstract Interpretation is **incomplete**: due to over-approximation, the bounds may be too loose to prove a true property, leading to a "don't know" state where the explanation is valid, but the verifier cannot prove it.

## B PROOF

THE ASYMMETRY OF PARALLEL FEATURE SELECTION

**Proposition B.1** (**Simultaneous Addition**). Any number of essential features can be added to the explanation **simultaneously**. This property allows us to leverage solvers capable of assessing *multiple verification queries in parallel*, leading to a substantial reduction in runtime.

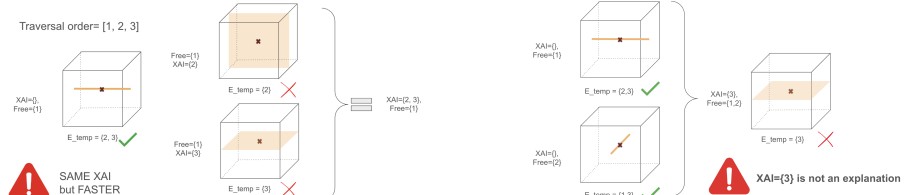

(a) Adding several features at once is sound.     (b) Freeing several features at once is unsound.

Figure 3: Toy example illustrating the asymmetry between adding and freeing features.

*Simultaneous Addition B.1.* Let $\mathcal{X}$ be the current explanation candidate, and let $\mathcal{R} = \{r_1, \ldots, r_k\}$ be a set of features not in $\mathcal{X}$. If, for every $r_i \in \mathcal{R}$, removing the single feature $r_i$ from the set $\mathcal{F} \setminus (\mathcal{X} \cup \{r_i\})$ produces a counterexample, then all features in $R$ are necessary and can be added to the explanation at once. □

*Simultaneous freeing 4.1.* If removing any feature from a set $\mathcal{R} \subseteq \mathcal{F} \setminus \mathcal{X}$ individually causes the explanation to fail (i.e., produces a counterexample), then all features in $\mathcal{R}$ can be added to the explanation $\mathcal{X}$ simultaneously. □

*Batch-Certifiable Freeing 4.2.* For any $i \neq c$ and $x' \in \Omega(x)$, lirpa bounds give $f_i(x') - f_c(x') \leq \overline{b}^i(x) + \sum_{j \in \mathcal{A}} \Delta_{i,j}(x')$ with $\Delta_{i,j}(x') \leq c_{i,j}$. Taking the worst case over $x'$ and $i$ yields $f_i(x') - f_c(x') \leq \Phi(\mathcal{A}) \leq 0$, precluding a label flip.

□

———

PROPOSITION (CORRECTNESS OF THE RECURSIVE PROCEDURE)

Let $\mathcal{A}$ be the set returned by Algorithm 2 augmented with the final singleton refinement step that tests each remaining feature individually with the LiRPA certificate $\Phi(\cdot)$. Then:

(i) (No singleton extension) For every feature $j \in \mathcal{F} \setminus \mathcal{A}$ we have

$$\Phi(\mathcal{A} \cup \{j\}) > 0,$$

i.e. no single feature can be added to $\mathcal{A}$ while preserving the certificate. Hence $\mathcal{A}$ is *singleton-maximal* with respect to the LiRPA certificate.

(ii) (Termination) Algorithm 2 terminates in at most $|\mathcal{F}|$ outer iterations (and finitely many inner steps).

(iii) (Full abstract minimality — conditional) If the inner batch solver called by Algorithm 2 returns, for each tested budget $p$, a *globally optimal* certified free set (i.e., for the current domain it finds a maximum-cardinality $\mathcal{A}_p$ satisfying $\Phi(\mathcal{A}_p) \leq 0$), then the final $\mathcal{A}$ is a globally maximal certified free set: there is no $\mathcal{A}' \supsetneq \mathcal{A}$ with $\Phi(\mathcal{A}') \leq 0$. In this case $\mathcal{A}$ is a true minimal abstract explanation (with respect to the chosen LiRPA relaxation).

**Proof.** **(i) No singleton extension.** By construction, the algorithm performs a final singleton refinement: it tests every feature $j \in \mathcal{F} \setminus \mathcal{A}$ by evaluating the certificate on $\mathcal{A} \cup \{j\}$. The algorithm only adds $j$ to $\mathcal{A}$ if $\Phi(\mathcal{A} \cup \{j\}) \leq 0$. Since the refinement ends with no further additions, it follows that for every remaining $j$ we have $\Phi(\mathcal{A} \cup \{j\}) > 0$. This is exactly the stated property.

**(ii) Termination.** Each time the algorithm adds at least one feature to $\mathcal{A}$, the cardinality $|\mathcal{A}|$ strictly increases and cannot exceed $|\mathcal{F}|$. The outer loop therefore performs at most $|\mathcal{F}|$ successful additions. If an outer iteration yields no new features, the loop stops. Inner loops (scanning budgets $p$ or performing singleton checks) are finite since they iterate over finite sets. Hence the algorithm terminates in finite time.

**(iii) Full abstract minimality under optimal inner solver.** Suppose that for every domain tested, the inner routine (called for each $p$) returns a certified free set of maximum possible cardinality among all subsets that satisfy $\Phi(\cdot) \leq 0$ on that domain. During each outer iteration the algorithm enumerates budgets $p$ (or otherwise explores the space of allowed cardinalities) and selects the largest $\mathcal{A}_p$ found; then $\mathcal{A}$ is augmented by that largest globally-feasible batch. If no nonempty globally-feasible batch exists for any tested $p$, then no superset of the current $\mathcal{A}$ can be certified (because any superset would have some cardinality $p'$ tested and the solver would have returned it). After the final singleton checks (which also use the optimal verifier on singletons), there remains no single feature that can be added. Combining these facts yields that no superset of $\mathcal{A}$ is certifiable, i.e. $\mathcal{A}$ is a globally maximal certified free set, as claimed. $\square$

**Abstract Minimal Explanation**

*Correctness of Iterative Singleton Freeing.* Let $\mathcal{F}$ be the candidate feature set and let $\mathcal{A}_0 \subseteq \mathcal{F}$ be an initial free set such that the LiRPA certificate verifies $\mathcal{A}_0$ (i.e. $\Phi(\mathcal{A}_0) \leq 0$). Run the Iterative Singleton Freeing procedure (Algorithm 5) with traversal order $\pi$. The algorithm returns a set $\mathcal{A}$ with the following properties:

1. **(Soundness)** The final set $\mathcal{A}$ satisfies $\Phi(\mathcal{A}) \leq 0$ (every added singleton was certified).

2. **(Termination)** The algorithm terminates after at most $|\mathcal{F}| - |\mathcal{A}_0|$ successful additions (hence in finite time).

3. **(Singleton-maximality)** For every $j \in \mathcal{F} \setminus \mathcal{A}$ we have $\Phi(\mathcal{A} \cup \{j\}) > 0$, i.e. no remaining single feature can be certified as free. $\square$

*Proof.* **Soundness (invariant).** By assumption $\Phi(\mathcal{A}_0) \leq 0$. The algorithm only appends a feature $i$ to the current free set after a LiRPA call returns success on $\mathcal{A} \cup \{i\}$, i.e. $\Phi(\mathcal{A} \cup \{i\}) \leq 0$. Since LiRPA certificates are sound, every update preserves the invariant "current $\mathcal{A}$ is certified". Therefore the final $\mathcal{A}$ satisfies $\Phi(\mathcal{A}) \leq 0$.

**Termination.** Each successful iteration increases $|\mathcal{A}|$ by one and $|\mathcal{A}| \leq |\mathcal{F}|$. Thus there can be at most $|\mathcal{F}| - |\mathcal{A}_0|$ successful additions. The algorithm halts when a full scan yields no addition; since scans iterate over a finite set ordered by $\pi$, the procedure terminates in finite time.

**Singleton-maximality.** Assume by contradiction that after termination there exists $j \in \mathcal{F} \setminus \mathcal{A}$ with $\Phi(\mathcal{A} \cup \{j\}) \leq 0$. The final scan that caused termination necessarily tested $j$ (traversal order covers all remaining indices), so the algorithm would have added $j$, contradicting termination. Hence for every $j \in \mathcal{F} \setminus \mathcal{A}$ we must have $\Phi(\mathcal{A} \cup \{j\}) > 0$, proving singleton-maximality. $\square$

**Worked counterexample (illustrating joint freeing).** Consider a toy binary classifier with two input features $x_1, x_2$ and property $\mathcal{P}$: the label remains class 0 iff $f_0(x') - f_1(x') \geq 0$. Suppose the LiRPA relaxation yields conservative linear contributions such that

$$\overline{b} + c_1 > 0, \qquad \overline{b} + c_2 > 0, \quad \text{but} \quad \overline{b} + c_1 + c_2 \leq 0,$$

where $c_i$ is the worst-case contribution of feature $i$ and $\overline{b}$ is the baseline margin. Then neither singleton $\{1\}$ nor $\{2\}$ is certifiable (each violates the certificate), but the joint set $\{1, 2\}$ is certifiable. The iterative singleton procedure terminates without adding either feature, while a batch routine (or an optimal MKP solver) would free both. This demonstrates the algorithm's limitation: it guarantees only singleton-maximality, not global maximality over multi-feature batches.

**Complexity and practical cost.** Let $n = |\mathcal{F}|$. In the worst case the algorithm may attempt a LiRPA call for every remaining feature on each outer iteration. If $r$ features are eventually added, the total number of LiRPA calls is bounded by

$$(n) + (n-1) + \cdots + (n-r+1) \;=\; r \cdot n - \frac{r(r-1)}{2} \;\leq\; \frac{n(n+1)}{2} = \mathcal{O}(n^2).$$

Thus worst-case LiRPA call complexity is quadratic in $n$. In practice, however, each successful addition reduces the candidate set and often many iterations terminate early; empirical behavior tends to be much closer to linear in $n$ for structured data because (i) many features are certified in early passes and (ii) LiRPA calls are highly parallelizable across features and can exploit GPU acceleration. Finally, the dominant runtime factor is the per-call cost of LiRPA (forward/backward bound propagation); therefore hybrid strategies (batch pre-filtering, prioritized traversal orders, occasional exact-solver checks on promising subsets) are useful to reduce the number of expensive LiRPA evaluations.

FAME FOR DISCRETE DATA

FAME, as presented, uses LiRPA, which is designed for continuous ( ) domains. A discrete feature $j$ with admissible values in a finite set $S_j$ can be incorporated by specifying an interval domain, which is the standard abstraction used in LiRPA-based verification.

Consequently, FAME allows a discrete feature to vary over its admissible values. LiRPA supports this by assigning

$$x'_j \in [\min S_j, \; \max S_j],$$

or, if only a subset $S'_j \subseteq S_j$ is permitted,

$$x'_j \in [\min S'_j, \; \max S'_j],$$

provided that the values form a contiguous range.

If a feature belongs to the explanation, it is fixed to its nominal value, which corresponds to assigning the zero-width interval $[x_j, x_j]$.

Note that freeing a feature to a non-contiguous set (e.g., allowing $\{1, 4\}$ but excluding $\{2, 3\}$) cannot be represented exactly, since LiRPA abstractions are convex intervals. Extending LiRPA to arbitrary finite non-convex domains is left for future work. In practice, such cases are rare: when categorical

values have no meaningful numeric ordering, one-hot encodings are standard, and each coordinate becomes a binary $\{0,1\}$ feature naturally supported by interval domains.

Since FAME only requires sound per-feature lower and upper bounds, all its components, including the batch certificate $\Phi(A)$ and the refinement steps, apply directly to discrete and categorical features.

## C  ALGORITHM

This appendix details the algorithmic procedures supporting the FAME framework and its baselines. We present four key algorithms:

- **Algorithm 3 (BINARYSEARCH)**: An enhanced version of the binary search traversal strategy used in Verix+. It employs a divide-and-conquer approach to identify irrelevant features, accepting a generic verification oracle (e.g., Marabou or LiRPA) as an input parameter.

- **Algorithm 4 (Simultaneous Add)**: An acceleration heuristic that uses adversarial attacks to quickly identify necessary features. By checking if relaxing a specific feature immediately leads to a counterexample via attacks (e.g., PGD), we can efficiently add necessary features to the explanation without expensive verification calls.

- **Algorithm 5 (Iterative Singleton Freeing)**: A refinement procedure that iterates sequentially through the remaining candidate features. It utilizes LiRPA certificates to check if individual features can be safely freed, serving as a final cleanup step for features that could not be certified in batches.

- **Algorithm 5 (Recursive Abstract Batch Freeing)**: The core recursive loop of our framework. It iteratively tightens the perturbation domain using cardinality constraints (varying $m$) and invokes the greedy batch-freeing heuristic to maximize the size of the abstract explanation, concluding with a singleton refinement step.

### C.1  VERIX+

In this enhanced BINARYSEARCH algorithm, the solver (e.g., Marabou or Lirpa) is passed as an explicit parameter to enable the CHECK function, which performs the core verification queries.

---

**Algorithm 3** BINARYSEARCH($f$, $x_\Theta$, solver)

---

1:  **function** BINARYSEARCH($f$, $x_\Theta$, solver)
2:     **if** $|x_\Theta| = 1$ **then**
3:        **if** CHECK($f$, $x_B \cup x_\Theta$, solver) **then**
4:           $x_B \leftarrow x_B \cup x_\Theta$
5:           **return**
6:        **else**
7:           $x_A \leftarrow x_A \cup x_\Theta$
8:           **return**
9:        **end if**
10:    **end if**
11:    $x_\Phi, x_\Psi = \text{split}(x_\Theta, 2)$
12:    **if** CHECK($f$, $x_B \cup x_\Phi$, solver) **then**
13:       $x_B \leftarrow x_B \cup x_\Phi$
14:       **if** CHECK($f$, $x_B \cup x_\Psi$, solver) **then**
15:          $x_B \leftarrow x_B \cup x_\Psi$
16:       **else**
17:          **if** $|x_\Psi| = 1$ **then**
18:             $x_A \leftarrow x_A \cup x_\Psi$
19:          **else**
20:             BINARYSEARCH($f$, $x_\Psi$, solver)
21:          **end if**
22:       **end if**
23:    **else**
24:       **if** $|x_\Phi| = 1$ **then**
25:          $x_A \leftarrow x_A \cup x_\Phi$
26:       **else**
27:          BINARYSEARCH($f$, $x_\Phi$, solver)
28:       **end if**
29:    **end if**
30: **end function**

---

## C.2   SIMULTANEOUS ADD

---

**Algorithm 4** Simultaneous Add

---

1:  **Input:** model $f$, input $x$, candidate set $\mathcal{F}$, current free set $\mathcal{A}$, adversarial procedure ATTACK(,) property $\mathcal{P}$
2:  **Initialize:** $\mathcal{E} \leftarrow \emptyset$                   ▷ set of necessary features
3:  **for** $i \in \mathcal{F} \setminus \mathcal{A}$ **do**
4:     $\mathcal{F}' \leftarrow \mathcal{F} \setminus \{i\}$
5:     **if** ATTACK($f$, $\Omega(x, \mathcal{F}')$, $\mathcal{P}$) succeeds **then**
6:        $\mathcal{E} \leftarrow \mathcal{E} \cup \{i\}$              ▷ $i$ must remain fixed
7:     **end if**
8:  **end for**
9:  **Return:** $\mathcal{E}$

---

## C.3 ITERATIVE SINGLETON FREEING

---

**Algorithm 5** Iterative Singleton Free

---

1: **Input:** model $f$, input $x$, candidate set $\mathcal{F}$, free set $\mathcal{A}$, certificate method L$_I$RPA(,) traversal order $\pi$, property $\mathcal{P}$
2: **repeat**
3:     found $\leftarrow$ **false**
4:     **for** $i \in \pi$ with $i \in \mathcal{F} \setminus \mathcal{A}$ **do**
5:         **if** L$_I$RPA$(f, \Omega(x, \mathcal{A} \cup \{i\}), \mathcal{P})$ succeeds **then**
6:             $\mathcal{A} \leftarrow \mathcal{A} \cup \{i\}$
7:             found $\leftarrow$ **true**
8:             **break**             ▷ restart scan from beginning of $\pi$
9:         **end if**
10:     **end for**
11: **until** found $=$ **false**
12: **Return:** $\mathcal{A}$

---

## C.4 RECURSIVE SIMULTANEOUS FREE

---

**Algorithm 6** Recursive Abstract Batch Freeing

---

1: **Input:** model $f$, input $x$, candidate set $\mathcal{F}$
2: **Initialize:** $\mathcal{A} \leftarrow \emptyset$             ▷ certified free set
3: **repeat**
4:     $\mathcal{A}_{best} \leftarrow \emptyset$
5:     **for** $m = 1 \ldots |\mathcal{F} \setminus \mathcal{A}|$ **do**
6:         $\mathcal{A}_m \leftarrow$ GREEDYABSTRACTBATCHFREEING$(f, \Omega^m(x; \mathcal{A}), \mathcal{F} \setminus \mathcal{A})$
7:         **if** $|\mathcal{A}_m| > |\mathcal{A}_{best}|$ **then**
8:             $\mathcal{A}_{best} \leftarrow \mathcal{A}_m$
9:         **end if**
10:     **end for**
11:     $\mathcal{A} \leftarrow \mathcal{A} \cup \mathcal{A}_{best}$
12: **until** $\mathcal{A}_{best} = \emptyset$
13: $\mathcal{A} =$ ITERATIVE SINGLETON FREE(f, x, $\mathcal{F}$, $\mathcal{A}$)     ▷ refine by testing remaining features
14: **Return:** $\mathcal{A}$

---

# D EXAMPLES

## D.1 ILLUSTRATION OF ABDUCTIVE EXPLANATION

Figure 4 illustrates a 3D classification task. For the starred sample, we seek an explanation for its classification within a local cube-shaped domain. As shown in Figure 5, fixing only feature $\mathbf{x}_2$ (i.e. freeing $\{\mathbf{x}_1, \mathbf{x}_3\}$, restricting perturbations to the orange plane) is not enough to guarantee the property, since a counterexample exists. However, fixing both $\mathbf{x}_2$ and $\mathbf{x}_3$ (orange line on free $x_1$) defines a 'safe' subdomain where the desired property holds true, since no counterexample exists in that subdomain. Therefore, $\mathcal{X} = \{\mathbf{x}_2, \mathbf{x}_3\}$ is an abductive explanation. Since neither $\{\mathbf{x}_2\}$ nor $\{\mathbf{x}_3\}$ are explanations on their own, $\{\mathbf{x}_2, \mathbf{x}_3\}$ is minimal. But it is not minimum since $\mathcal{X} = \{\mathbf{x}_1\}$ is also a minimal abductive explanation with a smaller cardinality. Two special cases are worth noting: an empty explanation (all features are irrelevant) and a full explanation (the entire input is necessary).

If all features are irrelevant, the explanation is the empty set, and no valid explanation exists. Conversely, if perturbing any feature in the input $\mathbf{x}$ changes the prediction, the entire input must be fixed, making the full feature set the explanation.

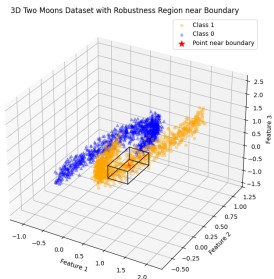

Figure 4: A 3D classification task.

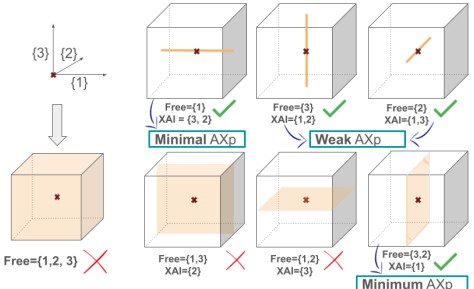

Figure 5: AXps with different properties.

## D.2 ILLUSTRATION OF THE KNAPSACK FORMULATION

This is an example demonstrating how the greedy heuristic described in Algorithm 1 works. Given a multi-class classification problem with three classes: 0, 1, and 2. The model correctly predicts class 0 for a given input. We want to free features from the irrelevant set $\mathcal{A}$ based on the abstract batch certificate. We have three candidate features to free: $j_1$, $j_2$, and $j_3$. The baseline budgets for the non-ground-truth classes are:

- Class 1: $-\bar{b}^1 = 10$
- Class 2: $-\bar{b}^2 = 20$

The normalized costs for each feature are calculated as $c_{i,j}/(-\bar{b}^i)$:

Table 2: Example of Greedy Heuristic Decision Making

| Feature | Normalized Cost for Class 1 | Normalized Cost for Class 2 | Maximum Normalized Cost |
|---------|------------------------------|------------------------------|--------------------------|
| $(j)$ | $(c_{1,j}/(-\bar{b}^1))$ | $(c_{2,j}/(-\bar{b}^2))$ | $(\max_i)$ |
| $j_1$ | $2/10 = 0.2$ | $8/20 = 0.4$ | 0.4 |
| $j_2$ | $7/10 = 0.7$ | $4/20 = 0.2$ | 0.7 |
| $j_3$ | $3/10 = 0.3$ | $3/20 = 0.15$ | 0.3 |

The algorithm's objective is to minimize the maximum normalized cost across all non-ground-truth classes. As shown in the table, the minimum value in the "Maximum Normalized Cost" column is 0.3, which corresponds to feature $j_3$. Therefore, the greedy heuristic selects feature $j_3$ to be added to the free set in this step, as it represents the safest choice.

## E EXPERIMENTS

### E.1 MODEL SPECIFICATION

We evaluated our framework on standard image benchmarks including the MNISTYann (2010) and GTSRBStallkamp et al. (2012) datasets. We used both fully connected and convolutional models trained in a prior state-of-the-art VERIX+Wu et al. (2024b) to perform our analysis.

The MNIST dataset consists of $28 \times 28 \times 1$ grayscale handwritten images. The architectures of the fully connected and convolutional neural networks trained on this dataset are detailed in Table 3 and Table 4, respectively. These models achieved prediction accuracies of 93.76% for the fully connected model and 96.29% for the convolutional model.

The GTSRB dataset contains colored images of traffic signs with a shape of 32×32×3 and includes 43 distinct categories. In the models used for our experiments, which were trained by the authors of VERIX+, only the 10 most frequent categories were used to mitigate potential distribution shift and obtain higher prediction accuracies. The architectures of the fully connected and convolutional

Table 3: Architecture of the MNIST-FC model.

| Layer Type | Input Shape | Output Shape | Activation |
|---|---|---|---|
| Flatten | $28 \times 28 \times 1$ | 784 | - |
| Fully Connected | 784 | 10 | ReLU |
| Fully Connected | 10 | 10 | ReLU |
| Output | 10 | 10 | - |

Table 4: Architecture of the MNIST-CNN model.

| Layer Type | Input Shape | Output Shape | Activation |
|---|---|---|---|
| Convolution 2D | $28 \times 28 \times 1$ | $13 \times 13 \times 4$ | - |
| Convolution 2D | $13 \times 13 \times 4$ | $6 \times 6 \times 4$ | - |
| Flatten | $6 \times 6 \times 4$ | 144 | - |
| Fully Connected | 144 | 20 | ReLU |
| Output | 20 | 10 | - |

models trained on GTSRB are presented in Table 5 and Table 6, respectively. These networks achieved prediction accuracies of 85.93% and 90.32%, respectively.

Table 5: Architecture of the GTSRB-FC model.

| Layer Type | Input Shape | Output Shape | Activation |
|---|---|---|---|
| Flatten | $32 \times 32 \times 3$ | 3072 | - |
| Fully Connected | 3072 | 10 | ReLU |
| Fully Connected | 10 | 10 | ReLU |
| Output | 10 | 10 | - |

Table 6: Architecture of the GTSRB-CNN model.

| Layer Type | Input Shape | Output Shape | Activation |
|---|---|---|---|
| Convolution 2D | $32 \times 32 \times 3$ | $15 \times 15 \times 4$ | - |
| Convolution 2D | $15 \times 15 \times 4$ | $7 \times 7 \times 4$ | - |
| Flatten | $7 \times 7 \times 4$ | 196 | - |
| Fully Connected | 196 | 20 | ReLU |
| Output | 20 | 10 | - |

The CIFAR-10 dataset contains colored images of common objects with a shape of $32 \times 32 \times 3$ and includes 10 distinct categories. The architecture of the ResNet-2B model used is detailed in Table 7. This model (sourced from the Neural Network Verification Competition (VNN-COMP) Wang et al. (2021)) is a compact residual network benchmark designed for neural network verification on CIFAR-10. Intended to help verification tools evolve beyond simple feedforward networks, this model was adversarially trained with an $L_\infty$ perturbation epsilon of $2/255$.

Table 7: Architecture of the ResNet-2B model (CIFAR-10).

| Layer Type | Input Shape | Output Shape | Activation |
|---|---|---|---|
| Reshape | 3072 | $32 \times 32 \times 3$ | - |
| Convolution 2D | $32 \times 32 \times 3$ | $16 \times 16 \times 8$ | ReLU |
| Residual Block (Downsample) | $16 \times 16 \times 8$ | $8 \times 8 \times 16$ | ReLU |
| Residual Block | $8 \times 8 \times 16$ | $8 \times 8 \times 16$ | ReLU |
| Flatten | $8 \times 8 \times 16$ | 1024 | - |
| Fully Connected | 1024 | 100 | ReLU |
| Output | 100 | 10 | - |

### E.2 DETAILED EXPERIMENTAL SETUP

We configured the VERIX+ implementation with the following settings: binary_search=true, logit_ranking=true, and traversal_order=bounds. To identify necessary features, we used the Fast Gradient Sign (FGS) technique for singleton attack addition, though the Projected Gradient Descent (PGD) is also available for this purpose.

We performed a comprehensive sensitivity analysis covering: (1) Solver Choice: Table 1 shows the Greedy heuristic finds explanations nearly identical in size to the optimal MILP solver (gap < 9 features), validating its near-optimality. (2) Cardinality Constraints: Figure 4 confirms that using the constraint (card=True) yields significantly smaller explanations. (3) Perturbation Magnitude ($\epsilon$): While we adhered to standard baseline values used by the baseline VERIX+ (e.g., 0.05 for MNIST, 0.01 for GTSRB) to ensure a direct and fair comparison, we acknowledge that explanation size is inversely related to $\epsilon$, as larger radii result in looser bounds.

### E.3 SUPPLEMENTARY EXPERIMENTAL RESULTS

**PERFORMANCE WITH ITERATIVE REFINEMENT** The three plots compare the performance of a greedy heuristic with an exact MILP solver for an iterative refinement task. The central finding across all three visualizations is that the greedy heuristic provides a strong trade-off between speed and solution quality, making it a more practical approach for large-scale problems.

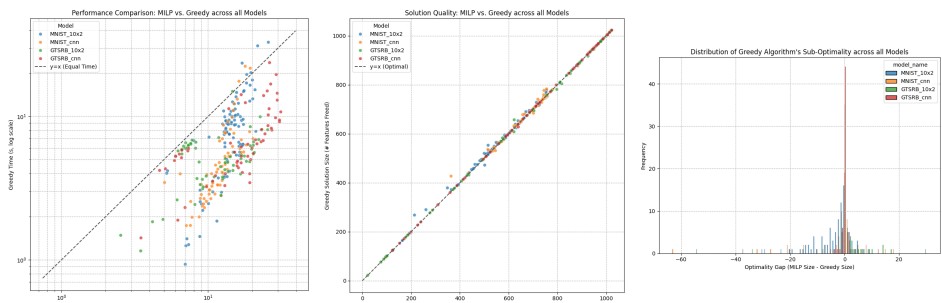

Figure 6: **Performance Comparison of FAME's Abstract Batch Freeing Methods.** These three plots compare the greedy heuristic against the exact MILP solver for the **iterative refinement** task for all the models. The first plot shows the runtime comparison of the two methods on a log-log scale. The second plot compares the size of the freed feature set for both methods. The third plot illustrates the distribution of the optimality gap (MILP size - Greedy size).

**Analysis of FAME's Abstract Batch Freeing** The visualizations demonstrate that the greedy heuristic provides a strong trade-off between speed and solution quality for the iterative refinement task.

- **Runtime Performance:** As shown in the first plot, the greedy algorithm is consistently faster than the MILP solver. This is evidenced by the data points for all models lying significantly below the diagonal line, confirming a substantial gain in runtime.

- **Solution Quality:** The second plot shows that the greedy algorithm produces solutions of comparable quality to the optimal MILP solver. The tight clustering of data points along the diagonal line for all models indicates a strong correlation between the sizes of the freed feature sets.

- **Optimality Gap:** The histogram of the final plot reinforces these findings by showing that the greedy heuristic frequently achieves the optimal solution, with the highest frequency of samples occurring at a gap of zero. The distribution further confirms that any sub-optimality is typically minimal.

# F  SCALABILITY ANALYSIS ON COMPLEX ARCHITECTURES (RESNET-2B ON CIFAR-10)

To validate the scalability of FAME on architectures significantly deeper and more complex than standard benchmarks, we conducted an evaluation on the ResNet-2B model (2 residual blocks, 5 convolutional layers, 2 linear layers) trained on the CIFAR-10 dataset Wang et al. (2021). We utilized an $L_\infty$ perturbation budget of $\epsilon = 2/255$. These additional experiments were conducted on a server equipped with an NVIDIA A100 80GB GPU.

**Feature Definition.**  For these experiments, we define the feature set $\mathcal{F}$ at the **pixel level**. Consequently, the total number of features is $N = 32 \times 32 = 1024$. Freeing a feature in this context corresponds to simultaneously relaxing the constraints on all three color channels (RGB) for that specific pixel location.

**Feasibility and Comparison.**  Running exact formal explanation methods (such as the complete VERIX+ pipeline with Marabou) on this architecture resulted in consistent timeouts or memory exhaustion, confirming that exact minimality is currently out of reach for this complexity class. In contrast, FAME successfully terminated for all processed samples.

**Detailed Quantitative Results by Configuration.**  To rigorously assess the contribution of each component in the FAME framework, we evaluated three configurations ($N = 100$ samples). The results are summarized below:

- **Single-Round Abstract Freeing (Algorithm 1 only).** This baseline represents a static approach without domain refinement.
    - *Performance:* It freed an average of only **5.53 features** (pixels).
    - *Insight:* This confirms that on deep networks, the initial abstract bounds are too loose to certify meaningful batches in a single pass. A static traversal strategy would fail here.
    - *Solver Comparison:* The Greedy heuristic (5.53 features, 50.8s) performed identically to the optimal MILP solver (5.37 features, 50.8s), validating the heuristic's quality.
- **Recursive Abstract Refinement (Algorithm 5).** This configuration enables the iterative tightening of the domain $\Omega^m(x; \mathcal{A})$.
    - *Performance:* The average number of freed features jumped to **476.38 pixels** (approx 46% of the image).
    - *Insight:* This dramatic increase (from ~5 to ~477) proves that the adaptive abstraction mechanism is critical. By iteratively constraining the cardinality, FAME recovers features that were previously masked by over-approximation.
    - *Solver Comparison:* Remarkably, even in this complex iterative setting, the Greedy approach (size 476.38) remained extremely close to the optimal MILP solution (size 477.76), with a negligible gap of $< 0.4\%$. This strongly justifies using the faster Greedy heuristic for scalability.
    - *Runtime:* The average runtime for this intensive recursive search was approximately **1934.94 seconds** (~32 minutes).
- **Full Pipeline (Iteration + Singleton Refinement).** This represents the final output of the complete FAME pipeline, including final safety checks and singleton refinement.
    - *Explanation Compactness:* The pipeline successfully certified a robust explanation with an average of **240.84 freed features** (pixels) across the full dataset.
    - *Efficiency:* The breakdown confirms that FAME can navigate the search space of deep networks where exact enumerations fail, producing sound abstract explanations ($WAXp^A$) significantly faster than the timeout threshold of exact solvers.

**Discussion and Future Directions.**  While the computational cost (~32 mins) is higher than for smaller models, these results establish that the Abstract Batch Certificate ($\Phi$) and recursive refinement scale mathematically to residual connections without theoretical blockers. The gap between

the abstract explanation size and the true minimal explanation is driven primarily by the looseness of the abstract bounds (LiRPA CROWN) on deep networks. Future work integrating tighter abstract interpretation methods (e.g., $\alpha$-CROWN) into the FAME engine will directly improve these results.

## G    DISCLOSURE: USAGE OF LLMS

An LLM was used solely as a writing assistant to correct grammar, fix typos, and enhance clarity. It played no role in generating research ideas, designing the study, analyzing data, or interpreting results; all of these tasks were carried out exclusively by the authors.