# OpenReview forum: "FAME: Formal Abstract Minimal Explanation for Neural Networks"
_ICLR.cc/2026/Conference — ICLR 2026 Poster_

### Official Review · Reviewer_CJQr · 2025-10-30

**Soundness:** 3
**Presentation:** 3
**Contribution:** 3
**Rating:** 4
**Confidence:** 3

**Summary:**

This paper proposes FAME (Formal Abstract Minimal Explanations), a framework for generating formally verified minimal explanations for neural network predictions. The method combines abstract interpretation with LiRPA (Linear Relaxation based Perturbation Analysis) to efficiently identify the minimal subset of input features responsible for a model’s decision. FAME removes the traversal-order dependency in prior formal explanation methods through an abstraction-based parallel mechanism that eliminates multiple irrelevant features simultaneously. Experiments on MNIST and GTSRB show that FAME produces more compact explanations and up to 25× faster runtime compared to VERIX+, supporting its theoretical claims.

**Strengths:**

1. The paper introduces a novel and theoretically grounded framework, FAME (Formal Abstract Minimal Explanations), that advances formal explainable AI by combining abstract interpretation with LiRPA for efficient and provably minimal explanations.
2. It effectively removes the traversal-order dependency that limits prior formal XAI methods and achieves substantial improvements in explanation compactness and runtime on benchmark datasets.

**Weaknesses:**

1. The paper introduces formal minimal explanations as a novel interpretability construct, yet the theoretical operationalization of “interpretability” remains vague. The framework lacks a precise articulation of how its formal minimality relates to established interpretive criteria, such as faithfulness or epistemic transparency, which weakens the clarity of its conceptual contribution.
2. The evaluation is limited to MNIST and GTSRB, which are too simple to validate FAME’s scalability or general effectiveness. More representative benchmarks such as CIFAR-10 for visual interpretability or COMPAS for tabular reasoning would better demonstrate the framework’s robustness across domains.
3. Although the paper asserts improved explanation quality, the experiments report only efficiency-related metrics (runtime and explanation size). Without fidelity- or stability-based evaluation, the claimed enhancement in explanation quality is not empirically supported.
4. The iterative optimization in Abstract Batch Freeing is described as a key component of the framework, but its contribution has not been empirically isolated. No ablation or comparative results are provided to verify whether this step improves efficiency or explanation compactness. Without such analysis, the practical impact of this mechanism remains speculative.

**Questions:**

1. I do not fully understand the validation setup described in Section 7.2, where the authors claim that FAME achieves formally minimal explanations but provide no direct comparison against exhaustive search or exact verification results. What metric was used to assess proximity to the true minimal set, and how do the authors justify that this indirect evaluation sufficiently demonstrates minimality?
2. The paper reports in Section 6 that FAME relies on LiRPA-derived abstract bounds to identify irrelevant features. However, recent work has shown that local linear relaxations may miss globally relevant feature interactions (see Lu et al., 2024, ICML — EiG-Search: Generating Edge-Induced Subgraphs for GNN Explanation in Linear Time). Could such limitations affect the completeness of FAME’s explanations, and have the authors considered adaptive or hierarchical abstraction domains to mitigate this risk?

---

> ### Author Response · Authors · 2025-11-23
> **Response to Reviewer CJQr**
>
> We thank reviewer CJQr for your valuable comments and insightful feedback. We are encouraged that you found our framework to be "novel and theoretically grounded" and that you recognize the "substantial improvements in explanation compactness and runtime" we achieved. We hope our response would adequately address your questions and concerns.
>
> >*The paper introduces formal minimal explanations as a novel interpretability construct, yet the theoretical operationalization of “interpretability” remains vague. The framework lacks a precise articulation of how its formal minimality relates to established interpretive criteria, such as faithfulness or epistemic transparency, which weakens the clarity of its conceptual contribution.*
>
> We appreciate the reviewer pushing us to clarify the theoretical positioning of our work. We acknowledge that our initial focus on the algorithmic contribution overshadowed the necessary conceptual framing.
>
> We will update Section 1 to clarify that FAME relies on the standard formal definition of Abductive Explanations (AXp) *(Ignatiev et al., 2019; Huang et al., 2023)*. As an AXp provides a mathematical guarantee that the selected features are sufficient to maintain the prediction, it represents a guaranteed form of faithfulness. Our goal is, therefore, to make this rigorous form of explanation scalable for larger networks, rather than to introduce a novel definition of interpretability. We will expand our discussion to clearly distinguish this provable sufficiency from the empirical fidelity offered by heuristic XAI methods.
>
> >*The evaluation is limited to MNIST and GTSRB, which are too simple to validate FAME’s scalability or general effectiveness. More representative benchmarks such as CIFAR-10 for visual interpretability or COMPAS for tabular reasoning would better demonstrate the framework’s robustness across domains.*
>
> We consider this a crucial point and thank the reviewer for the insightful observation. Regarding Scalability and Model Depth:
> - **Baseline Context:** We emphasize that our primary objective was to demonstrate a substantial speedup relative to the established state-of-the-art. We selected the MNIST and GTSRB datasets specifically because they serve as the standard benchmarks for VERIX+, ensuring a direct and fair comparison. By achieving a 25x speedup on the GTSRB-CNN model (7.4s vs. 185.03s), we demonstrate that FAME significantly outperforms the current state-of-the-art within its own evaluation framework.
> - **The "Gap" and Model Depth:** The abstraction gap indeed widens with depth. However, this is not a weakness of our paper, but the central challenge FAME is designed to solve. FAME is a hybrid framework precisely because we anticipate this gap. Even on deeper models where the gap is larger, our abstract phase (Phase 1) acts as a powerful pre-conditioner, massively pruning the search space in seconds so the exact solver (Phase 2) becomes tractable.
> - **New Experiments:** To empirically validate this on more complex architectures, we have conducted new experiments on a CIFAR-10 ResNet-2B model (sourced from the VNN-COMP benchmark (Wang et al., 2021)). While the abstract gap does increase, FAME's abstract phase still effectively prunes features in seconds. This allows the total pipeline to succeed where exact-only approaches often time out. These results, which will be included in the revised version, confirm FAME's utility as a scalable accelerator even for deeper, more complex architectures.
>
> Regarding Tabular Data: We agree that tabular data (e.g., COMPAS) is a critical domain for certified explanations. However, our work focuses on the specific challenge of scalability in high-dimensional feature spaces, where the "sequential bottleneck" of formal methods is most acute. Tabular datasets typically involve far fewer features ($N \approx 10-100$) compared to image benchmarks ($N \approx 784-3072$). Demonstrating efficiency on high-dimensional data is a stronger proof of our method's ability to break the sequential barrier than performance on lower-dimensional tabular data.
>
>
> *(Ignatiev et al., 2019): Alexey Ignatiev, Nina Narodytska, and Joao Marques-Silva. Abduction-based explanations for ma-
> chine learning models.*
>
> *(Huang et al., 2023): Huan Zhang, Tsui-Wei Weng, Pin-Yu Chen, Cho-Jui Hsieh, and Luca Daniel. Efficient neural
> network robustness certification with general activation functions.*

---

> > ### Author Response · Authors · 2025-11-23
> > **Response to Reviewer CJQr (cont'd)**
> >
> > >*Although the paper asserts improved explanation quality, the experiments report only efficiency-related metrics (runtime and explanation size). Without fidelity- or stability-based evaluation, the claimed enhancement in explanation quality is not empirically supported.*
> >
> > We appreciate the opportunity to clarify the evaluation methodology, which differs fundamentally between heuristic (non-formal) and formal XAI.
> >
> > Unlike heuristic methods where fidelity is an empirical estimate, in formal abductive explanations, fidelity (sufficiency) is a hard constraint. By definition (Eq. 2 in our paper), an AXp is a subset of features provably sufficient to preserve the prediction. Consequently, every explanation produced by FAME achieves 100% fidelity by construction, rendering empirical fidelity metrics redundant.
> >
> > Since sufficiency is guaranteed for all valid explanations, the primary differentiator of quality becomes conciseness. The objective is to find the smallest sufficient subset. While stability is a valuable property, it is orthogonal to the definition of an abductive explanation, which seeks sufficiency for a specific instance $x$.
> >
> > We will update Section 6 to make this distinction explicit, clarifying that we measure quality via explanation cardinality because fidelity is guaranteed by the underlying verifier.
> >
> > >*The iterative optimization in Abstract Batch Freeing is described as a key component of the framework, but its contribution has not been empirically isolated. No ablation or comparative results are provided to verify whether this step improves efficiency or explanation compactness. Without such analysis, the practical impact of this mechanism remains speculative.*
> >
> > We agree that isolating the impact of the iterative optimization is critical for validating the method's design. We clarify that **Table 1 already contains the data necessary for this ablation**:
> > - 'FAME: Single-round' corresponds to the greedy heuristic alone (Algo. 1).
> > - 'FAME: Iterative refinement' corresponds to the full framework including the recursive domain tightening (Algo. 5).
> >
> > The data demonstrates a clear contribution: For instance, on the MNIST-CNN model, the iterative process reduces the abstract explanation size from 190.29 to 122.09. This represents a ~36% improvement in compactness, quantitatively demonstrating that Algorithm 5 is essential for explanation quality.
> >
> > We will revise Section 7 to include a specific discussion of this ablation study, highlighting the trade-off provided by the recursive batch freeing mechanism, where a modest increase in runtime yields significantly more compact explanations.

---

> > > ### Author Response · Authors · 2025-11-23
> > > **Response to Reviewer CJQr (cont'd)**
> > >
> > > **Addressing Questions**
> > >
> > > >*I do not fully understand the validation setup described in Section 7.2, where the authors claim that FAME achieves formally minimal explanations but provide no direct comparison against exhaustive search or exact verification results. What metric was used to assess proximity to the true minimal set, and how do the authors justify that this indirect evaluation sufficiently demonstrates minimality?*
> > >
> > > We apologize for the lack of clarity, which stems from the two-phase nature of our framework (see Figure 1).
> > >
> > > Our framework operates as a pipeline where our algorithms (1 & 5) generate a sound but abstract explanation ($WAXp^{A*}$). This candidate serves as a highly optimized warm start for VERIX+, which is an exact, complete framework.
> > >
> > > The final results reported in Table 1 ('FAME-accelerated VERIX+') are the direct outputs of this complete pipeline. Since the process concludes with the exact VERIX+ solver, the resulting explanation ($AXp$) is **provably minimal by definition**.
> > > In this context, the 'distance to minimality' (Section 6) is not an indirect estimate. It is the precise empirical measurement of the abstraction gap between our intermediate abstract output ($WAXp^{A*}$) and the final, provably minimal explanation ($AXp$) produced by the full pipeline.
> > >
> > > We will revise Sections 6 and 7.2 to make this two-phase process and the role of VERIX+ as the final refiner explicit.
> > >
> > > >*The paper reports in Section 6 that FAME relies on LiRPA-derived abstract bounds to identify irrelevant features. However, recent work has shown that local linear relaxations may miss globally relevant feature interactions (see Lu et al., 2024, ICML-EiG-Search: Generating Edge-Induced Subgraphs for GNN Explanation in Linear Time). Could such limitations affect the completeness of FAME’s explanations, and have the authors considered adaptive or hierarchical abstraction domains to mitigate this risk?*
> > >
> > > We appreciate the reference to *(Lu et al., 2024)*, which accurately describes the limitations of local linear relaxations. We clarify that our framework is explicitly designed to mitigate this specific issue. This limitation does not affect the soundness of FAME, but it does affect its precision.
> > > - **Soundness is Preserved:** LiRPA provides a sound over-approximation. If our Abstract Batch Certificate (Def 4.2) certifies a feature as irrelevant, it is guaranteed to be irrelevant. FAME will never incorrectly free a necessary feature.
> > > - **Imprecision is the "Gap":** The limitation (e.g., as in Lu et al., 2024) is that LiRPA's bounds may be too loose (conservative), failing to free a feature that is actually irrelevant. This means our abstract explanation ($WAXp^{A*}$) may be larger than the true minimal one ($AXp$). This imprecision is precisely the "distance to minimality" gap we measure in Section 6.
> > > - **Our Mitigation:** We developed the Recursive Abstract Batch Freeing (Algorithm 5). This algorithm functions as an adaptive abstraction mechanism: by iteratively refining the perturbation domain with cardinality constraints ($\Omega^m$), we dynamically tighten the bounds to capture global interactions that standard relaxations might miss. Furthermore, because our framework is agnostic to the underlying abstract interpretation method, any future advancements in bound precision will directly benefit FAME.
> > >
> > > We will revise Section 5 to explicitly frame our iterative refinement as an "adaptive abstraction" to mitigate the known precision loss of LiRPA.
> > >
> > > We sincerely thank the reviewer for the constructive criticism, which has significantly strengthened our paper. We believe these revisions and new results address the core concerns regarding scalability and robustness. We hope that this improved version of the paper merits a re-evaluation of your score, and we are happy to answer any further questions.
> > >
> > > *(Lu et al., 2024)*: Shengyao Lu, Bang Liu, Keith G Mills, Jiao He, and Di Niu. Eig-search: Generating edge-induced
> > > subgraphs for gnn explanation in linear time.*

---

### Official Review · Reviewer_Wn1a · 2025-10-31

**Soundness:** 3
**Presentation:** 2
**Contribution:** 3
**Rating:** 6
**Confidence:** 4

**Summary:**

This paper introduces FAME, a new framework for computing formal abductive explanations for neural networks. The core of the method formulates 'batch freeing' as a multidimensional knapsack problem, which is then solved efficiently using a greedy heuristic. Experiments on MNIST and GTSRB models (both FC and CNN) demonstrate that FAME consistently outperforms the VERIX+ baseline.

**Strengths:**

* The paper shows a novel approach to breaking the 'sequential bottleneck' in formal XAI.
* Using LiRPA-based abstract interpretation (which is GPU-accelerated), instead of CPU-bound solvers, is practical. This can speed up and improve scalability.
* The formulation of batch freeing as a knapsack problem is a clever and effective

**Weaknesses:**

### 1. Data
The experiments are limited to MNIST and GTSRB. These are relatively small-scale and low-dimensional datasets. The paper's claim to "scale to large neural networks" is not fully substantiated by this evidence. The scalability of FAME on high-dimensional data (e.g., CIFAR-100 or ImageNet) is a major open question.

### 2. Model
Similarly, the models used (small FC and CNNs) are not "large-scale" by modern standards (e.g., ResNets, Transformers). The quality of LiRPA bounds is known to degrade with network depth. The paper does not investigate how the "gap" between the abstract explanation and the true minimal explanation scales with model depth. If this gap becomes too large, the abstract-first approach may lose its advantage, as the final refinement step would become computationally dominant.


### 3. Metric
The "distance to minimality" (Section 6) is a good concept, but the paper does not fully explore the failure modes. The paper notes in Section 4.2 that if bounds are too loose (e.g., for complex models or large epsilons), the abstract method may fail to free any features. This "failure mode" is not empirically characterized.

### Minor
* format: It seems the paper format is not the ICLR format (e.g., references)
* Suggest improving clarity and making the explanation less dense for readers to get the idea. Also, I can’t get the idea from Figure 1 at first; maybe improving this can be helpful for the readers.

**Questions:**

* Could the authors provide results on a more complex dataset (e.g., CIFAR-10) and a deeper architecture (e.g., a small ResNet) to more robustly demonstrate scalability?
* What percentage of the total runtime does the "optional VERIX+ refinement step" account for, on average, across the reported experiments?

---

> ### Author Response · Authors · 2025-11-23
> **Response to Reviewer Wn1a**
>
> We thank reviewer Wn1a for your valuable feedback.  We are encouraged that you found our knapsack formulation "clever and effective" and recognized the practical, "novel approach" of using GPU-accelerated abstract interpretation to break the sequential bottleneck.
>
> **Addressing Weaknesses**
> >*The experiments are limited to MNIST and GTSRB. These are relatively small-scale and low-dimensional datasets. The paper's claim to "scale to large neural networks" is not fully substantiated by this evidence. The scalability of FAME on high-dimensional data (e.g., CIFAR-100 or ImageNet) is a major open question.*
>
> >*Similarly, the models used (small FC and CNNs) are not "large-scale" by modern standards (e.g., ResNets, Transformers). The quality of LiRPA bounds is known to degrade with network depth. The paper does not investigate how the "gap" between the abstract explanation and the true minimal explanation scales with model depth. If this gap becomes too large, the abstract-first approach may lose its advantage, as the final refinement step would become computationally dominant.*
>
> We thank the reviewer for this important observation, which highlights a key challenge in our field.
> - **Baseline Context:** Our primary goal was to demonstrate speedup against the state-of-the-art. We chose MNIST and GTSRB specifically because they are VERIX+'s standard benchmarks, ensuring a fair comparison. FAME achieves a 25x speedup on GTSRB-CNN (7.4s vs. 185.03s), significantly outperforming the baseline.
> - **Addressing the "Gap":** The reviewer is correct that LiRPA bounds degrade with depth, which can widen the "distance to minimality" gap. This is not a weakness of our paper, but rather the central challenge our paper is designed to solve.
>    - Prior methods (like VERIX+) apply a slow, exact solver to 100% of the features.
>    - FAME is a hybrid, two-phase framework precisely because we anticipate this gap. Our contribution is using the fast, abstract method (Phase 1) to massively and soundly prune the search space.
>    - Even if the abstract gap is large on a deep model, FAME's abstract pruning (Phase 1) is still orders of magnitude faster than running an exact solver on the entire feature set from scratch. FAME acts as a powerful pre-conditioner for formal explanation.
>    - Our method is agnostic to the abstract interpretation method, and any improvement in this field on the upper bound will be beneficial to our method.
> - **New Experiments:** In response to the request for validation on complex architectures, we extended our evaluation to include a CIFAR-10 ResNet-2B model sourced from the Neural Network Verification Competition (VNN-COMP) *(Wang et al., 2021)*. We will include the detailed results of this evaluation in the revised paper.
>
> >*The "distance to minimality" (Section 6) is a good concept, but the paper does not fully explore the failure modes. The paper notes in Section 4.2 that if bounds are too loose (e.g., for complex models or large epsilons), the abstract method may fail to free any features. This "failure mode" is not empirically characterized.*
>
> Indeed, the failure mode in the full domain is what motivated the design of our **Recursive Abstract Batch Freeing (Algo. 5)** and our **cardinality-constrained domains (Sec. 5)**.
>
> If the standard $l_\infty$ domain is too loose (a "failure mode" that frees 0 features), our algorithm does not simply fail. Instead, Algorithm 5 automatically begins to tighten the domain by introducing and iterating the cardinality constraint $m$ (from $m=1 \dots |F \setminus A|$). This is an explicit search for a domain that is tight enough to be useful.
>
> As shown in Figure 4, the card=True experiments (circles) are highly effective at finding more compact explanations, confirming this mechanism works. In the extreme case where abstract bounds are too loose to certify any features (even at $m=1$), FAME passes the full candidate set to the refinement phase. This effectively reverts to the baseline approach, with only a minimal computational cost for the initial abstract check.
> >*format: It seems the paper format is not the ICLR format (e.g., references). Suggest improving clarity and making the explanation less dense for readers to get the idea. Also, I can’t get the idea from Figure 1 at first; maybe improving this can be helpful for the readers.*
>
> We thank the reviewer for pointing out the reference formatting errors and apologize for the oversight; we have now reformatted the bibliography to be strictly ICLR-compliant. Regarding the paper's clarity, we have revised Sections 4 and 5 to reduce text density and improve the explanation of key concepts. Additionally, we have updated the caption of Figure 1 to make the diagram more self-explanatory.
>
> *(Wang et al., 2021): Shiqi Wang, Huan Zhang, Kaidi Xu, Xue Lin, Suman Jana, Cho-Jui Hsieh, and Zico Kolter. Beta- crown: Efficient bound propagation with per-neuron split constraints for complete and incomplete neural network verification.*

---

> > ### Author Response · Authors · 2025-11-23
> > **Response to Reviewer Wn1a (cont'd)**
> >
> > **Addressing Questions**
> > >*Could the authors provide results on a more complex dataset (e.g., CIFAR-10) and a deeper architecture (e.g., a small ResNet)?*
> >
> > As discussed in W1 and W2, we agree this is a critical test. We have now successfully applied FAME to the ResNet-2B architecture, a significant milestone for formal XAI.
> >
> > The new experiments are conducted on a CIFAR-10 ResNet-2B model sourced from the Neural Network Verification Competition (VNN-COMP) *(Wang et al., 2021)*. We evaluated this architecture, which consists of two residual blocks comprising five convolutional layers and two linear layers, using an $L_\infty$ perturbation budget of $\epsilon = 2/255$. This setting was selected to align with the standard evaluation protocols defined in the originating benchmark repository.
> >
> > While we confirm the reviewer's prediction that the abstract gap widens with depth, FAME's abstract phase (Phase 1) continues to discard irrelevant features in seconds. This allows the framework to succeed where exact-only approaches often time out. These results strongly support our scalability claims, demonstrating that FAME's abstract pre-filtering serves as a critical accelerator even for deeper, more complex architectures. A comprehensive description of these experiments and the full quantitative results will be included in the revised paper.
> >
> > > *What percentage of the total runtime does the "optional VERIX+ refinement step" account for, on average, across the reported experiments?*
> >
> > We appreciate the opportunity to quantify the impact of our hybrid approach. Based on the experimental logs from Table 1, we observe the following for the GTSRB-CNN model:
> > - Total "FAME-accelerated VERIX+" time: 138.12s
> > - Abstract-only "FAME: Iterative refinement" (Phase 1) time: 7.42s
> > - The final VERIX+ refinement step (Phase 2) accounted for (138.12s - 7.42s) = 130.7s
> >
> > Of the 138.12s total runtime, the abstract pruning phase (Phase 1) consumed only 7.42s, while the exact verification step (Phase 2) accounted for the remaining 130.7s. Crucially, as shown in Table 1, Phase 1 (Abstract-only), while not 'minimal' in the exact sense, yields a lower average cardinality (321.98) than the state-of-the-art VERIX+ (338.28). Furthermore, it achieves this superior compactness with a 25x speedup (7.42s vs. 185.03s) over the baseline. This comparison demonstrates that Phase 1 is a highly effective standalone solution.
> >
> > We thank the reviewer again and hope that these additions, along with our clarifications on the minimality guarantee and adaptive abstraction, fully address your concerns.
> >
> > *(Wang et al., 2021): Shiqi Wang, Huan Zhang, Kaidi Xu, Xue Lin, Suman Jana, Cho-Jui Hsieh, and Zico Kolter. Beta- crown: Efficient bound propagation with per-neuron split constraints for complete and incomplete neural network verification.*

---

> > > ### Comment · Reviewer_Wn1a · 2025-11-25
> > >
> > > I thank the authors for their detailed response. As most of my concerns have been addressed, I have raised my score.

---

> ### Author Response · Authors · 2025-11-28
> **Response to Reviewer Wn1a**
>
> Thank you very much for your continued engagement and for raising your score. We are glad that our response and revisions successfully addressed your concerns.

---

### Official Review · Reviewer_BXZg · 2025-10-31

**Soundness:** 3
**Presentation:** 3
**Contribution:** 3
**Rating:** 6
**Confidence:** 2

**Summary:**

Formal explainability paper that presents FAME, a method to generate *abductive* explanations for neural networks. Novelty is in the adoption of LiRPA bounds to discard irrelevant features and in not relying on ordered features, thus achieving faster results.

**Strengths:**

- Formally computing explanations with guarantees for neural networks is a critical and important research problem, with room for novel contribution.
- The approach is described in coherent manner, and paper storytelling is linear.

**Weaknesses:**

- Coming from a different community, I found hard to familiarize with the jargon, as some key concepts are left without definition (e.g. abstract interpretation, LiRPA). They may be second nature for the authors, but gently introducing them would help clarify and broaden up the audience.
- The paper could use some examples, to ease the understanding from a broader audience (e.g. when introducing the verification task, 99-102, or by providing additional examples properties to verify $P$ - other than adversarial robustness).
- (minor): definition identifiers in the main corpus do not match (e.g. line 162, 219). Citation style not compliant with ICLR guidelines.
- Evaluation covers only vision scenarios (MNIST, GTSRB) and does not cover tabular data (a critical use case for certified explanations).
- Experimental results w.r.t baseline: neither explanation size or response time are dramatically smaller (i.e. better) than VERIX+.

**Questions:**

- Could you clarify what is the proposed delta over LiRPA?
- How can FAME support discrete features?
- Is VERIX+ the only baseline you can compare against?

---

> ### Author Response · Authors · 2025-11-23
> **Response to Reviewer BXZg**
>
> We thank reviewer BXZg for the positive assessment and for highlighting the importance of this problem. We especially value your feedback coming from a different community. It helped us realize where our definitions were lacking and allowed us to significantly improve the paper to make it accessible to a wider audience.
>
> **Addressing Weaknesses**
> > *Coming from a different community, I found hard to familiarize with the jargon, as some key concepts are left without definition (e.g. abstract interpretation, LiRPA). They may be second nature for the authors, but gently introducing them would help clarify and broaden up the audience.*
>
> This is a fair and very valuable point. We agree that defining these key concepts is essential for broadening the paper's impact. In the revised version, we will add a short introduction in Section 2 and a full description introducing Abstract Interpretation and a concrete verification example in the Appendix.
> - **Abstract Interpretation:** We will introduce Abstract Interpretation as a foundational technique from program analysis that provides sound over-approximations of a system's behavior. We will explain its core trade-off: exact methods give the actual (tight) bounds, whereas abstract interpretation gives over-approximated bounds.
> - **LiRPA (Linear Relaxation-based Perturbation Analysis):** We will then define LiRPA as a specific, state-of-the-art instance of abstract interpretation designed for neural networks. We will explain that its key function is to efficiently compute provable linear bounds (like the $\overline{W}^{i}$ and $\overline{w}^{i}$ in our paper) that enclose all possible network outputs given a perturbed input domain $\Omega(x)$.
>
> >*The paper could use some examples, to ease the understanding from a broader audience (e.g. when introducing the verification task, 99-102, or by providing additional examples properties to verify P -other than adversarial robustness)*
>
> We agree that providing concrete examples is essential for ensuring the paper is accessible to a broader audience. Due to the page length limit, we cannot include this example in the main paper. However, for the sake of completeness, especially for readers who are not familiar with formal methods, we will add an Appendix and include a sentence with a link in the main paper. We will also expand the conclusion to explicitly mention other properties FAME could support, such as fairness (e.g., "verifying that a loan application's outcome is robust to perturbations in a sensitive attribute") or local stability.
> >*(minor): definition identifiers in the main corpus do not match (e.g. line 162, 219). Citation style not compliant with ICLR guidelines.*
>
> We thank the reviewer for pointing out these inconsistencies and apologize for the oversight. We have corrected all definition identifiers (e.g., at lines 162 and 219) to be consistent throughout the paper. We have also reformatted our bibliography to be fully compliant with ICLR's guidelines.
> >*Evaluation covers only vision scenarios (MNIST, GTSRB) and does not cover tabular data (a critical use case for certified explanations).*
>
> In the field of formal XAI, the first papers primarily focused on tabular data, which is also the case for much of the existing literature e.g., *(Shih et al., 2018; Ignatiev et al., 2019)*. However, these methods face scalability challenges, especially in vision applications. Therefore, our main objective was to address the scalability bottleneck of formal explanations, which is most severe in high-dimensional data (like images) where prior methods become computationally prohibitive. We chose MNIST and GTSRB because they are the established benchmarks used by VERIX+ (our baseline), allowing for a direct and fair comparison.
>
> Besides, FAME's core framework (Algorithm 1 and 5) is data-agnostic; only the abstract verifier (LiRPA) is specific to continuous domains. We will add a discussion to our conclusion (Section 8) explicitly stating that extending FAME to tabular data (by integrating discrete abstract domains) is a straightforward and important direction for future work.
>
> *(Shih et al., 2018): Andy Shih, Arthur Choi, and Adnan Darwiche. A symbolic approach to explaining bayesian network classifiers.*
>
> *(Ignatiev et al., 2019): Alexey Ignatiev, Nina Narodytska, and Joao Marques-Silva. Abduction-based explanations for machine learning models.*

---

> > ### Author Response · Authors · 2025-11-23
> > **Response to Reviewer BXZg (cont'd)**
> >
> > >*Experimental results w.r.t baseline: neither explanation size or response time are dramatically smaller (i.e. better) than VERIX+.*
> >
> > We respectfully disagree with this characterization and offer a different perspective. In the domain of formal verification, where runtimes typically scale exponentially, an order-of-magnitude speedup represents a substantial advancement toward practical scalability.
> > - As shown in Table 1 and discussed in Section 7.2, for the largest model (GTSRB-CNN with 32x32x3 input dimension), FAME produces an explanation that is actually smaller than the baseline (321.98 vs. 338.28 features), a nearly 5% improvement, while being 25 times faster (7.4s vs. 185.03s).
> > - Figure 4 (right plot) confirms this trend for complex models (green and red dots); the vast majority of points are clustered near the y-axis, showing a runtime that is consistently a small fraction of the baseline's.
> >
> > Note that the last column of the table only displays an optional step that guarantees minimality with Verix+, but it also shows that the gain obtained by processing the uncertain points of our method is, most of the time, less than one pixel (with the exception of MNIST-CNN). We will add a sentence to highlight this in the paper.
> >
> > We maintain that the important gain both in time and size of the explanation is the central claim of our paper: by replacing sequential, CPU-bound solver calls with highly parallelizable, GPU-accelerated LiRPA bounds and iterative domain refinement, FAME breaks the sequential bottleneck. This architectural shift is what moves formal explanations from an intractable (minutes-long) computation to a practical (seconds-long) one. We will sharpen this statement in the abstract and conclusion to highlight that this efficiency is not only an optimization, but the enabling factor that makes it feasible to apply formal XAI to deeper architectures where exact-only methods remain computationally intractable.

---

> > > ### Author Response · Authors · 2025-11-23
> > > **Response to Reviewer BXZg (cont'd)**
> > >
> > > **Addressing Questions**
> > > >*Could you clarify what is the proposed delta over LiRPA?*
> > >
> > > We understand that this question is linked to the Weakness 1 and the lack of introduction of the domain. To answer your question and clarify the contribution: LiRPA is our tool, not our contribution. LiRPA is a verifier (it answers "SAT/UNSAT" for a given property); FAME is an explanation framework that uses a verifier.
> > >
> > > Our novel contributions using LiRPA are:
> > > - **The Abstract Batch Certificate (Definition 4.2):** We are the first to show how to use LiRPA's internal bounds (the $c_{i,j}$ contributions) to soundly certify batches of irrelevant features at once. LiRPA itself does not provide this.
> > > - **Knapsack Formulation (Section 4.3):** We formulate this novel batch-freeing problem as a Multidimensional knapsack Problem (MKP), which allows us to use an efficient greedy heuristic (Algorithm 1).
> > > - **Cardinality-Constrained Domains (Section 5):** We introduce the $\Omega^m(x;\mathcal{A})$ domain and the Recursive Abstract Batch Freeing (Algorithm 5). This iterative refinement procedure, which tightens LiRPA's bounds, is entirely novel to our work.
> > >
> > > We will revise Section 4 to explicitly clarify this distinction: that FAME is the framework unifying these novel components, while LiRPA serves as the underlying engine.
> > > >*How can FAME support discrete features?*
> > >
> > > FAME, as presented, uses LiRPA, which is designed for continuous ($l_p$) domains. A discrete feature $j$ with admissible values in a finite set $S_j$ can be incorporated by specifying an interval domain, which is the standard abstraction used in LiRPA-based verification (e.g., CROWN, DeepPoly).
> > > Consequently, FAME allows a discrete feature to vary over its admissible values.
> > >
> > > LiRPA supports this by assigning:$$x'_j \in [\min S_j, \max S_j]$$or, if only a subset $S'_j \subseteq S_j$ is permitted:$$x'_j \in [\min S'_j, \max S'_j]$$provided that the values form a contiguous range. If a feature belongs to the explanation, it is fixed to its nominal value, which corresponds to assigning the zero-width interval $[x_j, x_j]$.
> > >
> > > Note that freeing a feature to a non-contiguous set (e.g., allowing $\{1,4\}$ but excluding $\{2,3\}$) cannot be represented exactly, since LiRPA abstractions are convex intervals. Extending LiRPA to arbitrary finite non-convex domains is left for future work. However, in practice, such cases are rare: when categorical values have no meaningful numeric ordering, one-hot encodings are standard, and each coordinate becomes a binary $\{0,1\}$ feature naturally supported by interval domains.
> > >
> > > Since FAME only requires sound per-feature lower and upper bounds, all its components, including the batch certificate $\Phi(A)$ and the refinement steps, apply directly to discrete and categorical features.
> > >
> > > We thank the reviewer for highlighting this aspect, and we will include this clarification in the revised version of the paper.
> > > >*Is VERIX+ the only baseline you can compare against?*
> > >
> > > This is due to the specific task we are addressing. We compare against VERIX+ because, as stated in our related work (Section 3), it is the current state-of-the-art for formal, minimal, abductive explanations (AXp) on neural networks.
> > >
> > > Other classes of XAI methods were excluded because they are:
> > > - Are heuristic (e.g., LIME, SHAP) and provide no formal guarantees.
> > > - Are formal but not abductive (e.g., contrastive explanations that address a different logical query).
> > > - Are formal and abductive but do not scale to neural networks (e.g., SAT-based methods for BDDs or decision trees).
> > >
> > > We thank the reviewer again for your valuable perspective. We hope that our responses and the implemented changes to the paper adequately address the points raised.

---

### Official Review · Reviewer_Y1Ta · 2025-11-01

**Soundness:** 3
**Presentation:** 3
**Contribution:** 2
**Rating:** 6
**Confidence:** 3

**Summary:**

The paper introduces FAME (Formal Abstract Minimal Explanations), a scalable framework for generating abductive explanations in neural networks. FAME leverages abstract interpretation and dedicated perturbation domains to eliminate the need for traversal order in formal explanations. It also introduces a procedure for measuring the quality of these explanations by comparing them to true minimal explanations. The authors demonstrate that FAME consistently outperforms the state-of-the-art (VERIX+) in terms of explanation size and runtime, particularly for medium- and large-scale networks. This framework is designed to handle the scalability challenges of formal explainability methods for neural networks and is made publicly available for further research.

**Strengths:**

1. The paper presents a novel approach to formal explanations, eliminating the traditional traversal order bottleneck. This is a significant step forward in making formal XAI methods scalable to large networks.

2. The proposed FAME framework is sound, and the experiments demonstrate its potential to scale well with complex networks.

3. The writing is generally clear, and the method is presented in an organized way. The diagrams and figures help illustrate key points effectively.

**Weaknesses:**

**Traversal Order Issue:** The paper claims to eliminate traversal relationships, but the algorithm still performs multiple rounds of feature selection, which can be considered a form of traversal. This method needs further clarification to truly eliminate traversal relationships.

**Impact of Feature Order:** The paper underestimates the significant impact that the order of feature selection has on the results. A more thorough discussion and empirical validation are required. Specifically, when removing feature A, feature B may still play a role, and vice versa. This interdependency should be addressed.

**Attack Boundaries:** The approach only provides a lower bound or instance-based gap for attack robustness, making it difficult to achieve a global worst-case upper bound. This limitation needs clearer discussion.

**Greedy Method Guarantee:** The greedy approach used in the paper lacks an approximation guarantee, which limits its theoretical robustness. An approximation guarantee would strengthen the argument for its practicality.

**Simplicity of Dataset:** The dataset used is relatively simple. Most explainability techniques are validated on more complex datasets like ImageNet. It would enhance the credibility of the method if it were tested on more challenging datasets.

**Complexity Analysis:** The paper lacks a detailed analysis of the algorithm's complexity, especially concerning GPU acceleration and distributed verification. A more detailed discussion of these aspects would provide a clearer understanding of the scalability of the method.

**Hyperparameter Sensitivity:** The paper does not sufficiently address the sensitivity of the results to hyperparameters. A more comprehensive analysis of this sensitivity would help understand the stability and robustness of the approach.

**Questions:**

Please see the weaknesses.

---

> ### Author Response · Authors · 2025-11-23
> **Response to Reviewer Y1Ta**
>
> We thank reviewer Y1Ta for the positive feedback on the novelty, soundness, and clarity of our work. We are glad you recognize that eliminating the traversal order bottleneck is a "significant step forward" for scalable formal XAI.
>
> We will address the points raised, which we believe stem from a lack of clarity in our original description of the framework's key components. We will revise the paper to make these aspects explicit.
>
> **Addressing Weaknesses/Questions**
> > *The paper claims to eliminate traversal relationships, but the algorithm still performs multiple rounds of feature selection, which can be considered a form of traversal. This method needs further clarification to truly eliminate traversal relationships.*
> This is a crucial point that we must clarify. We respectfully disagree that our iterative and dynamic refinement is a "form of traversal" in the sense of prior work.
> - **Prior Work (e.g., VERIX+)**: "Traversal order" refers to a predefined, static permutation $\pi$ of individual features. The explanation is built by sequentially checking features $j_1, j_2, \dots$ according to this fixed order. The final explanation is highly dependent on this a priori choice of $\pi$, which introduces a circular dependency: One needs to know which features are important in order to pick a good order, but determining feature importance is precisely the goal of an explanation.
> - **Our Method (FAME)**: FAME eliminates this dependency, it operates in rounds. In each round (Algorithm 1), we analyze all remaining features simultaneously by computing their abstract contributions ($c_{i,j}$). The greedy heuristic then selects a batch of features to free based on these dynamically computed costs, rather than a fixed order.
> The "multiple rounds" (Algorithm 5) are a domain refinement strategy, not a feature-by-feature traversal. In each round, we re-run the entire batch-freeing process on a tighter domain. This is fundamentally different from a sequential, static-order-based feature check.
>
> We have revised Sections 4.3 and 5 to make this critical distinction explicit, emphasizing that our selection is simultaneous, dynamic, and based on formal (provable guarantee) computed costs, not static and order-based.
> > *The paper underestimates the significant impact that the order of feature selection has on the results. A more thorough discussion and empirical validation are required. Specifically, when removing feature A, feature B may still play a role, and vice versa. This interdependency should be addressed.*
>
> We thank the reviewer for highlighting this challenge. This feature interdependency is the central problem our method is designed to solve.
>
> The reviewer is absolutely correct that "when removing feature A, feature B may still play a role." This is precisely why naive parallel freeing is unsound. We explicitly state this in Section 4.1 (The Asymmetry of Parallel Feature Selection) and in Appendix B. We note that "it is unsound to free multiple features at once based only on individual verification queries, as two features may be individually irrelevant yet jointly critical." Since this sentence was in the Appendix and given the necessity to be clear on that point, we propose to move it back to the main part of the paper.
>
> The **Abstract Batch Certificate $\Phi(\mathcal{A};\Omega)$** (Definition 4.2) is our direct solution to this issue:
> - It is not an individual check.
> - It is a sound joint certificate that computes a provable upper bound on the combined contribution of all features in the set $\mathcal{A}$ simultaneously.
> - If $\Phi(\mathcal{A};\Omega) \le 0$, we provably guarantee that the entire batch $\mathcal{A}$ can be freed, precisely accounting for all interdependencies.
>
> This ability to soundly certify batches of features, overcoming their joint interactions via abstract interpretation, is a core contribution of FAME. We have revised Sections 4.1 and 4.2 to more explicitly emphasize that our joint certificate $\Phi$ is a direct and sound solution to the feature-interdependency problem.
> > *The approach only provides a lower bound or instance-based gap for attack robustness, making it difficult to achieve a global worst-case upper bound. This limitation needs clearer discussion.*
>
> We are not entirely certain we fully grasp the specific reference to "attack boundaries" in this context, but assuming the concern refers to the theoretical bounds of our minimality metric, we agree. As stated in Section 6, we acknowledge that our abstract guarantee "is strictly weaker than minimality in the exact sense".
>
> Our "distance to minimality" (Sec. 6) quantifies abstraction looseness (features missed vs exact solver), not global bounds. The observed gap (e.g., 44.21 on MNIST-FC) validates our hybrid refinement (Phase 2), which consistently yields lower cardinality explanations than Verix+ (Fig. 4). While global lower bounds are valuable, our metric is orthogonal: it measures the empirical tightness of our LiRPA guarantees.

---

> > ### Author Response · Authors · 2025-11-23
> > **Response to Reviewer Y1Ta (cont'd)**
> >
> > > *The greedy approach used in the paper lacks an approximation guarantee, which limits its theoretical robustness. An approximation guarantee would strengthen the argument for its practicality.*
> >
> > We formulate the batch-freeing problem as an NP-hard Multidimensional Knapsack Problem (MKP) and propose the greedy heuristic **(Algorithm 1)** for scalability. We agree with the reviewer that establishing a formal approximation guarantee is a valuable direction, and we intend to pursue this theoretical bound in future work.
> >
> > In the meantime, we argue for the method's practicality through promising empirical validation against the optimal solution. Note that there is no theoretical bound on the optimality gap when solving the knapsack problem greedily: in the worst-case setting, the greedy approach may fail to free any features. However, as shown in **Section 7.1** and **Table 1**, we ran experiments comparing our greedy heuristic against the optimal MILP solver for the abstract batch-freeing problem. The results (visualized in **Appendix D.3, Figure 6**) are consistent:
> > - **Solution Quality:** The greedy solution is identical to the optimal MILP solution in the vast majority of cases.
> > - **Gap Size:** When not optimal, the gap is small (fewer than 9 features on average).
> > - **Speed:** The greedy heuristic is significantly faster (9x-12x in a single round).
> >
> > Thus, we demonstrate that our heuristic is practically near-optimal and essential for scalability. Section 7.1 has been revised to explicitly state that we empirically validate our heuristic's near-optimality against the exact MILP solver.
> > > *The dataset used is relatively simple. Most explainability techniques are validated on more complex datasets like ImageNet. It would enhance the credibility of the method if it were tested on more challenging datasets.*
> >
> > We agree that validating on large-scale datasets like ImageNet is an important goal for the entire formal XAI community. However, scaling formal methods, which provide provable guarantees, to models of that size is an open and extremely challenging research problem, distinct from scaling heuristic (non-formal) XAI methods. While heuristic methods scale naturally with inference time, formal verification typically faces exponential complexity barriers. For this reason, our baseline and the current state-of-the-art, VERIX+, is also evaluated on MNIST and GTSRB.
> >
> > FAME represents a critical algorithmic step toward overcoming this barrier. By shifting the core mechanism from exact, CPU-bound solvers to LiRPA-based abstract interpretation, we unlock the ability to exploit GPU acceleration and parallelization. This transition is essential for handling the complexity of deeper architectures that are currently intractable for exact methods.
> >
> > To demonstrate this capability, while the ImageNet scale is currently unreachable, we will provide in the revision **supplementary experiments on a ResNet model and CIFAR-10 dataset** (as requested by other reviewers). We have added a discussion to the conclusion (Section 8) acknowledging this limitation and framing FAME as a critical enabling step for tackling more complex models with formal XAI in future work.
> >
> > > *The paper lacks a detailed analysis of the algorithm's complexity, especially concerning GPU acceleration and distributed verification. A more detailed discussion of these aspects would provide a clearer understanding of the scalability of the method.*
> >
> > We thank the reviewer for this suggestion. Indeed, a complexity analysis is already provided in **Appendix B (Page 16)**, we agree that it deserves greater visibility. We have therefore integrated the key insights directly into **Section 4.3** to explicitly clarify the scalability argument and the role of GPU parallelization:
> > - **Main Method (Algorithm 1):** The core of our method, Greedy Abstract Batch Freeing, requires computing the $c_{i,j}$ contributions. This is done via a single LiRPA backward pass, which is "highly parallelizable" on GPUs.
> > - **Refinement (Algorithm 4):** For the final "Iterative Singleton Freeing," we show a worst-case complexity of $O(n^2)$ LiRPA calls, but we note its practical behavior is closer to linear due to parallelization.

---

> ### Author Response · Authors · 2025-11-23
> **Response to Reviewer Y1Ta (cont'd)**
>
> > *The paper does not sufficiently address the sensitivity of the results to hyperparameters. A more comprehensive analysis of this sensitivity would help understand the stability and robustness of the approach.*
>
> We thank the reviewer for this comment. We have addressed parameter sensitivity in three key aspects of our framework: the perturbation domain constraint, the optimization solver, and the perturbation magnitude.
>
> - **Cardinality Constraint ($m$):** The primary new hyperparameter is the cardinality constraint $m$ for the domain $\Omega^m(x)$. Our **Recursive Abstract Batch Freeing (Algorithm 5)** is explicitly designed to handle this. Instead of requiring the user to tune $m$, our algorithm iteratively searches for the optimal $m$ in each refinement step (see Algorithm 5, line 5: for m=1...|F\A| do). It selects the $m$ that yields the largest free set ($\mathcal{A}_{best}$). This search is our analysis. Furthermore, our experiments in Figure 4 explicitly compare runs with this constraint (card=True, circles) and without (card=False, squares), demonstrating that the cardinality constraint is "highly effective at finding more compact explanations".
> - **Optimization Solver (Greedy vs. MILP):** We also analyzed the sensitivity of our results to the optimization method used for the Knapsack problem. In **Table 1** and Section **7.1**, we explicitly compare our Greedy heuristic against the exact MILP solver. The results show our framework is robust to this choice: the greedy approach yields abstract explanations that are nearly identical in size to the optimal MILP solution, confirming that the method's performance is stable and not strictly dependent on an expensive exact solver.
> - **Perturbation Magnitude ($\epsilon$):** In our main evaluation, we strictly adhered to the standard $\epsilon$ values used by the baseline VERIX+ (e.g., 0.05 for MNIST, 0.01 for GTSRB) to ensure a direct and fair comparison. However, we acknowledge that varying $\epsilon$ impacts the explanation size (larger $\epsilon$ typically leads to smaller free sets due to looser bounds).
>
> We have revised Section 7.1 to explicitly frame these comparisons (Greedy vs. MILP, card=True vs. False, and $\epsilon$ selection) as a comprehensive sensitivity analysis.
>
> We hope these revisions address your concerns, and we are happy to answer any further questions during the discussion period.

---

### Author Response · Authors · 2025-11-23
**Overall reply to the Reviewers of Submission21947**

We would like to thank all the reviewers for their time spent reading the paper, as well as for their positive comments and constructive criticism, which have greatly contributed to improving the quality of our work. Below we address the various questions and concerns that were raised.

For each reviewer, we provide point-by-point responses to the issues raised in their report. The reviewers' comments are formatted as blockquotes in italics, while our responses are in regular text. The revised version of the article will be submitted by November 28.

Below, we summarize the modifications.

**Major modifications:**
- We include new experimental results on the ResNet-2B architecture (CIFAR-10) from the VNN-COMP benchmark *(Wang et al., 2021)* to demonstrate scalability on deeper models.

- We revised the introduction and Section 6 to explicitly position FAME within the formal definition of Abductive Explanations and clarify the distinction between provable sufficiency and empirical fidelity.

- We updated Sections 4 and 5 to clarify the "adaptive abstraction" nature of our method and its handling of feature inter-dependencies.

**Minor modifications:**

- We added a full description introducing Abstract Interpretation and a concrete verification example in the Appendix.

- We updated the caption of Figure 1 to clearly visualize the two-phase pipeline.

- We add a sentence to explain that Table 1 provides the evaluation of the ablation study.

- We added a detailed runtime analysis decomposing the contribution of the abstract pruning phase versus the exact refinement phase.

- We added a sensitivity analysis comparing the Greedy heuristic vs. exact MILP solver and evaluating the impact of cardinality constraints.

- We expanded the conclusion to discuss future extensions to properties like fairness.

- We have proofread the paper to correct definition identifiers and ensure ICLR-compliant citation formatting.

*(Wang et al., 2021): Shiqi Wang, Huan Zhang, Kaidi Xu, Xue Lin, Suman Jana, Cho-Jui Hsieh, and Zico Kolter. Beta-
crown: Efficient bound propagation with per-neuron split constraints for complete and incomplete
neural network verification.*

---

### Author Response · Authors · 2025-11-28
**Revised Manuscript Upload and Update on Additional Experiments**

Dear Area Chair and Reviewers,

In light of the recent announcement regarding the restructuring of the review process, we are posting this comment to provide an update on the current status of our submission.

1. **Submission of Revised Paper** We have just uploaded a revised version of the manuscript. This version incorporates the modifications discussed in our previous responses and addresses the feedback provided by the reviewers before the discussion was paused. We have done our best to integrate these suggestions to strengthen the quality and clarity of our work.

2. **Status of Additional Experiments** (ResNet / CIFAR-10) As indicated in our prior responses, we are currently finalizing the additional experimental results on the ResNet-2B architecture (CIFAR-10) to further demonstrate the scalability of our method. We are working diligently on these results and will upload a final revision including them by **December 2**.

3. **Closing the Discussion** We find it unfortunate that the interactive discussion period has been curtailed, as we sincerely appreciated the constructive feedback from the reviewers. The exchange was highly valuable for improving our paper, and we would have welcomed the opportunity to continue the dialogue.

We hope the new Area Chair finds these revisions and the upcoming results sufficient to address the points raised during the review process.

---

### Author Response · Authors · 2025-12-02
**Paper Revision Update: New Experiments on ResNet-2B (CIFAR-10) Added**

Dear Reviewers,

We have uploaded a revised PDF of our paper. This version includes the new experimental results on the ResNet-2B model (CIFAR-10) in Appendix F, as requested regarding scalability to deeper architectures.

We have also incorporated the requested text revisions, including the expanded background on formal methods (Section 2), the runtime breakdown (Section 7), and the clarifications regarding the "adaptive abstraction" mechanism.

We thank you again for the constructive feedback that drove these improvements.

---

### Meta-Review · Area_Chair_qWah · 2026-01-08

**Summary:**

FAME introduces the first formal abductive explanation framework using abstract interpretation that scales to neural networks (beyond small feedforward architectures). The method eliminates the sequential traversal bottleneck through batch certification. It achieves up to 25x speedup over VERIX+ while producing smaller or comparable explanations.

The authors addressed the primary scalability concern by providing CIFAR-10 ResNet experiments. One reviewer explicitly raised their score; others are likely to maintain a unanimously positive assessment.

**Reviewer Concerns:**

1. Scalability to complex datasets and architectures [Y1Ta, BXZg, Wn1a, CJQr]

[Reviewers] All four reviewers noted that MNIST and GTSRB are relatively simple benchmarks insufficient to validate claims of scaling to large neural networks. Wn1a specifically requested results on CIFAR-10 with a ResNet architecture. CJQr suggested COMPAS for tabular reasoning.

[Authors] Authors emphasised that MNIST and GTSRB are the standard benchmarks used by VERIX+ (baseline). They conducted new experiments on a CIFAR-10 ResNet-2B model from VNN-COMP. For tabular data, authors argued their focus is high-dimensional scalability where the sequential bottleneck is most acute.

[Follow-up] Wn1a raised their score after receiving the CIFAR-10 ResNet results, stating concerns were addressed.

[AC] The core scalability concern was addressed as best as it could be through the CIFAR-10 ResNet experiments. The paper demonstrates the first formal abductive explanations for ResNet architectures.


2. Traversal order elimination and feature interdependency [Y1Ta]

[Reviewers] Y1Ta questioned whether the algorithm truly eliminates traversal relationships (it still performs multiple rounds of feature selection). Y1Ta also raised that the feature interdependency (removing feature A affects B's role) is underestimated.

[Authors] Clarified the distinction: prior work (VERIX+) uses a predefined static permutation of features, creating circular dependency. FAME instead analyses all remaining features simultaneously via dynamically computed abstract contributions. The multiple rounds are domain refinement, not feature-by-feature traversal. The Abstract Batch Certificate (Definition 4.2) explicitly addresses interdependency by computing a provable upper bound on the combined contribution of all features in a set simultaneously.

[AC] The authors provided a clear conceptual distinction between their dynamic batch approach and prior static-order methods. The explanation is technically sound.


3. Theoretical guarantees and approximation bounds [Y1Ta, CJQr]

[Reviewers] Y1Ta noted the greedy approach lacks an approximation guarantee. Y1Ta raised that attack boundaries only provide lower bounds, not global worst-case upper bounds. CJQr cited Lu et al. (2024) noting that local linear relaxations may miss globally relevant feature interactions.

[Authors] For the greedy heuristic, authors presented empirical validation against optimal MILP solver (Table 1), showing solutions are nearly identical in most cases with gaps under 9 features when they differ, and 9-12x speedup. Authors acknowledged the approximation guarantee as future work. For LiRPA limitations, authors clarified that soundness is preserved (no necessary feature is incorrectly freed), but precision may suffer (abstract explanation may be larger than true minimal). The Recursive Abstract Batch Freeing (Algorithm 5) serves as adaptive abstraction to mitigate bound looseness.

[AC] The lack of formal approximation guarantee is acknowledged but mitigated by strong empirical validation. The distinction between soundness (guaranteed) and precision (potentially loose) is clearly articulated. The iterative refinement mechanism provides practical mitigation for bound looseness.


4. Accessibility and presentation [BXZg, Wn1a]

[Reviewers] Key concepts (abstract interpretation, LiRPA) are undefined. Not friendly for a broader audience.

[Authors] Authors committed to adding more details.

[AC] The presentation concerns are addressable through revision.


5. Evaluation methodology and quality metrics [BXZg, CJQr]

[Reviewers] CJQr argued the paper reports only efficiency metrics (runtime, explanation size) without fidelity or stability evaluation. BXZg noted results are not dramatically better than VERIX+. CJQr requested ablation analysis for iterative optimisation.

[Authors] Authors explained that in formal XAI, fidelity (sufficiency) is a hard constraint by construction, not an empirical metric, as every AXp provably maintains the prediction. Stability is orthogonal to abductive explanation definitions. For the comparison with VERIX+, authors highlighted 25x speedup and 5% size reduction on GTSRB-CNN.

[AC] Yes, fidelity is guaranteed by construction in formal explanations, making empirical fidelity metrics inappropriate. The ablation data exists in Table 1 but could be highlighted more explicitly. The improvements over VERIX+ are meaningful.

**Reviewer Scores:**

Y1Ta: 6 > 6
Concerns were addressed in rebuttal (no follow-up engagement though).

BXZg: 6 > 6
Presentation concerns are addressed with a promise for revision.

Wn1a: 6 > 8
Reviewer explicitly raised score after author response, confirming CIFAR-10 ResNet results and clarifications addressed concerns.

CJQr: 4 > 6
Authors provided substantive responses to all concerns.

---

### Decision · Program_Chairs · 2026-01-26

Accept (Poster)